# Emission or atmospheric processes? An attempt to attribute the source of large bias of aerosols in eastern China simulated by global climate models

Tianyi Fan[1], Xiaohong Liu[1,2], Po-Lun Ma[3], Qiang Zhang[4], Zhanqing Li[1,5], Yiquan Jiang[6], Fang Zhang[1],
Chuanfeng Zhao[1], Xin Yang[1], Fang Wu[1], Yuying Wang[1]

[1]College of Global Change and Earth System Science, State Key Laboratory of Earth Surface Processes and Resource Ecology, and Joint Center for Global Change and Green China Development, Beijing Normal University, Beijing, China
[2]Department of Atmospheric Science, University of Wyoming, Laramie, Wyoming, USA
[3]Atmospheric Sciences and Global Change Division, Pacific Northwest National Laboratory, Richland, Washington, USA
[4]Center for Earth System Science, Tsinghua University, Beijing, China
[5]Department of Atmospheric and Oceanic Science & ESSIC, University of Maryland, College Park, Maryland, USA
[6]Institute for Climate and Global Change Research, School of Atmospheric Sciences, Nanjing University, Nanjing, China

*Correspondence to*: Tianyi Fan (fantianyi@bnu.edu.cn) and Xiaohong Liu (xliu6@uwyo.edu)

**Abstract.** Global climate models often underestimate aerosol loadings in China and these biases can have significant implications for anthropogenic aerosol radiative forcing and climate effects. The biases may be caused either by the emission inventory or the treatment of aerosol processes in the models, or both, but so far no consensus has been reached. In this study, a relatively new emission inventory based on energy-statistics and technology, Multi-resolution Emission Inventory for China (MEIC), is used to drive the Community Atmosphere Model version 5 (CAM5) to evaluate aerosol distribution and radiative effects against observations in China. The model results are compared with the model simulations with the widely used IPCC AR5 emission inventory. We find that the new MEIC emission improves the aerosol optical depth (AOD) simulations in eastern China and explains 22%-28% of the AOD low bias simulated with the AR5 emission. However, AOD is still low biased in eastern China. Seasonal variation of the MEIC emission leads to a better agreement with the observed seasonal variation of primary aerosols than the AR5 emission, but the concentrations are still underestimated. This implies that the atmospheric loadings of primary aerosols are closely related to the emission, which may still be underestimated over eastern China. In contrast, the seasonal variations of secondary aerosols depend more on aerosol processes (e.g., gas and aqueous phase production from precursor gases) that are associated with meteorological conditions and to a less extent on the emission. It indicates that the emissions of precursor gases for the secondary aerosols alone cannot explain the low bias in the model. Aerosol secondary production processes in CAM5 should also be revisited. The aerosol-meteorology interaction can also influence the gas- and aqueous-phase chemistry and aerosol loadings. However, the local changes in temperature and relative humidity are small since wind fields are prescribed. The simulation using MEIC estimates the annual averaged aerosol direct radiative effects (ADREs) at the top of atmosphere (TOA), surface, and atmosphere to be -5.02, -18.47, and 13.45 W m$^{-2}$ respectively over eastern China, which are enhanced by -0.91, -3.48, and 2.57 W m$^{-2}$

compared with the AR5 emission. The differences of ADREs by using MEIC and AR5 emissions are larger than the decadal changes of the modeled ADREs, indicating the uncertainty of the emission inventories. This study highlights the importance

of improving both the emission and aerosol secondary production processes in modeling the atmospheric aerosols and their radiative effects. Yet, if the estimations of MEIC emissions in trace gases do not suffer similar biases as in the AOD, our findings would help affirm a fundamental error in the conversion from precursor gases to secondary aerosols as hinted in other recent studies following different approaches.

Keywords: Emission inventory in China; Aerosol processes in GCMs; Aerosol direct radiative effects; CAM5

## 1 Introduction

As indicated by previous studies, many global climate models (GCMs) suffer from substantially low biases of aerosol loadings in East Asia, in particular, the rapidly developing region of eastern China. Nearly all GCMs that participate in the Atmospheric Chemistry and Climate Model Intercomparison Project (ACCMIP, Lamarque et al., 2013) have low bias of the aerosol optical depth (AOD) in East Asia by about -36% to -58% compared with Aerosol Robotic Network (AERONET) observations (Shindell et al., 2013). The AOD biases are substantially larger than those in North America and Europe. The low biases of aerosol loadings can have significant implications for anthropogenic aerosol radiative forcing and climate effects (Boucher et al., 2013; Myhre et al., 2013). It also suggests that the aerosol forcing and climate effects assessed by Intergovernmental Panel on Climate Change (IPCC) could be much underestimated due to the large aerosol biases in China (Liao et al., 2015).

Anthropogenic emissions of aerosols and precursor gases are hypothesized to be one of the leading-order reasons for the large simulation error (Liu et al., 2012). China has been experiencing three decades of rapid economic growth that brings emissions of atmospheric pollutants that are very different from the past and other parts of the world (Streets et al., 2008; Zhang et al., 2009; Klimont et al., 2009; Lu et al., 2011; Lei et al., 2011; Wang et al., 2012). Nowadays China is a large contributor to global aerosol emissions (Liao et al., 2015) and radiative forcing (B. Li et al., 2016). However, the emission inventory in China remains highly uncertain due to limited knowledge of the rapid changing economy and the variety of technologies in production, energy-use, and emission control (Zhao et al., 2011; Fu et al., 2012; Wang F. et al., 2014; Chang et al., 2015; Zhang et al., 2015). When used as input to the model simulations, the emission inventories can significantly affect the model output of aerosol concentrations and their radiative effects. It is estimated that the uncertainties of simulated surface concentrations of different aerosol species due to emission range from 3.9% to 40.0% over eastern China (Chang et al., 2015). Model experiments show that moderate (20%-30%) adjustments of regional emissions exert considerable influence on global AOD and aerosol radiative forcing (Yu et al., 2013; He and Zhang, 2014). It is noteworthy that the ACCMIP models, most of which underestimate the AOD in East Asia (Shindell et al., 2013), have different treatments of aerosol processes but use the same IPCC Fifth Assessment Report (AR5) emission inventory (Lamarque et al., 2010). This implies that the IPCC AR5 emission inventory may underestimate the emission in East Asia. Unique features of the anthropogenic emissions in China include the elevated level of sulfate and black carbon (BC) emissions in the winter heating season in northern China and high level of $NO_x$ and $NH_3$ emissions that are linked to the winter haze in recent years.

On the other hand, the treatments of aerosol processes can also bring bias in the model. In the real world, aerosols originate from direct emissions of primary particles (e.g., sea salt, dust, primary organics, BC, and a small fraction of sulfate) or secondary particles formed from precursor gases (e.g., sulfate, nitrate, secondary organics). After emission, the precursor gases experience gas and aqueous phase transformation to form the secondary aerosols. A newly emitted or formed aerosol particle will go through a series of atmospheric processes (e.g., condensational growth, coagulation with another particle, transport, water uptake, wet scavenging/cloud processing, and dry deposition) until it completes its life cycle in the

atmosphere. The inter-model diversity of global aerosol burden and optical properties largely depend on the treatment of

aerosol processes in each individual model and to a less extent on the differences of the emissions among models (Textor et al., 2007). Modifications of the gas-phase chemistry and inorganic aerosol treatment in the Community Atmospheric Model version 5 (CAM5) improve the model performance for aerosol mass and AOD (He and Zhang, 2014), but substantial low biases still exist for East Asia. Most GCMs including CAM5 do not include the aqueous phase chemistry on preexisting particles, which proves to be important for the formation of the winter hazes in northern China (Wang et al., 2013; He et al.,

2014; Huang et al., 2014; Wang X.Y. et al. 2014; Wang Y.S. et al., 2014; Zheng et al, 2015; Chen et al., 2016; Dong et al., 2016; Wang et al., 2016; Cheng et al., 2016). The aerosol impacts on meteorological fields and clouds further affect the aerosol pollution condition in the lower troposphere. The aerosol radiative effects induced by BC or other aerosol components could stabilize boundary layer and thus reduce the height of boundary layer, tending to exacerbate aerosol pollution near the ground (Wang et al., 2013). Specifically, the aging of black carbon (BC) considerably enhances light

absorption (Khalizov et al., 2009; Peng et al., 2016). The aerosol-cloud interaction might modify temperature and moisture profiles and precipitation (Wang et al., 2011), leading to potential feedback on the atmospheric chemistry. With all the above mentioned uncertainties mingled in the GCMs, it is not clear whether the emission or the aerosol processes are more responsible for the low biases of AOD simulated by GCMs in eastern China.

In this study we attempt to understand the attribution of the low biases of AOD in eastern China simulated by GCMs. First,

we examine the effect of changing the anthropogenic emission of China in a global climate model (i.e., CAM5) on improving the aerosol simulation. CAM5 significantly underestimates AOD in East Asia [*Liu et al.* 2012] and the normalized mean bias of AOD is one of the largest among the ACCMIP models investigated in Shindell et al. (2013). We compare the aerosol simulation in CAM5 using the default IPCC AR5 emission inventory with the simulation using a new one that better represents the magnitude and seasonal variation of the emissions. Second, with the inclusion of seasonality in the new

emission inventory, we attempt to isolate the impacts of aerosol processes on the seasonal variation of aerosol concentrations from the impact of emission. Aerosol processes that depend on the meteorological factors (*e.g.,* temperature, humidity, wind speed, etc.) in the model are analyzed to explain the impact of emission on the secondary aerosols versus the impact of emission on the primary aerosols. Finally, we examine the impact of the uncertainty of the emission inventories on the aerosol direct radiative effects (ADREs). The differences of ADREs due to the use of the two emission inventories are

calculated and compared with the change of ADREs in the last decade due to the change of emission in China.

This paper is organized as follows. Section 2 describes the model setup, the emission inventories, and the observations. Section 3 shows the results of aerosol properties and ADREs simulated by CAM5 using the new MEIC emission compared to the AR5 emission and analyzes the impacts of emission and aerosol processes. Section 4 discusses the uncertainty of the emission inventories by comparing with the decadal changes of ADREs due to emission change. Conclusions are provided in

Section 5.

## 2 Method

### 2.1 Model setup and experiments

We run CAM5 with the 3-mode Modal Aerosol Model (MAM3), which prognoses aerosol mass/number size distribution and mixing state in the Aitken, accumulation, and coarse modes (Liu et al., 2012). The simulated primary aerosol species include BC, primary organic matter (POM), sea salt, and dust, while the secondary aerosol species include sulfate and secondary organic aerosol (SOA). The aerosol species are assumed to be internally mixed within modes and externally mixed among modes. The physical, chemical, and optical properties of aerosols are simulated in a physically based manner. Aerosol processes include transport, gas and aqueous phase (in cloud water only) chemical reactions for sulphur species, microphysics (nucleation, condensational growth, and coagulation), dry deposition, wet scavenging, and water uptake. Efficient secondary formation of aerosol in Beijing, China has been reported, characterized by frequent nucleation events preceding the pollution episodes followed by rapid condensational growth during the episodes (Qiu et al., 2013; Guo et al., 2014; Zhang et al., 2015). For treatment of these processes in CAM5, a binary $H_2SO_4$-$H_2O$ homogeneous nucleation scheme (Vehkamaki et al., 2002) is used and a cluster activation scheme (Shito et al., 2006) is applied in the planetary boundary layer. Condensation of $H_2SO_4$ vapor and semi-volatile organics to the aerosol modes is treated dynamically using the mass transfer expressions (Seinfeld and Pandis, 1998) that are integrated over the size distribution of each mode (Binkowski and Shankar, 1995). Coagulation between Aitken and accumulation modes is considered. Water uptake is based on the equilibrium Köhler theory (Ghan and Zaveri, 2007). SOA formation is based on fixed mass yields, i.e., the percentage of semi-volatile organic compounds (VOCs) that could form SOA, with one additional step of complexity by explicitly simulating the emission and condensation/evaporation of the condensable organic vapours (i.e., the lumped semi-volatile organic gas species, SOAG) that are generated from VOCs. The aerosol optical properties are parameterized by Ghan and Zaveri (2007). The refractive indices for most aerosol components are taken from OPAC (Hess et al., 1998), but for BC the value (1.95, 0.79i) from Bond and Bergstrom (2006) is used. More details of the aerosol treatments can be found in Liu et al. (2012).

We conduct two CAM5 simulations with different anthropogenic emission inventories in China for year 2009. The first simulation uses the emission inventory that follows the protocol of the IPCC AR5 experiments (the AR5 emission inventory hereinafter, see Lamarque et al., 2010). The second simulation is driven by an improved technology-based Multi-resolution Emission Inventory for China (MEIC) developed at Tshinghua University (the MEIC emission inventory hereinafter, http://www.meicmodel.org/index.html). MEIC has the following advantages: (1) adoption of a detailed technology-based approach, (2) application of a dynamic methodology of rapid technology renewal, (3) re-examination of China's energy statistics, and (4) monthly emissions to represent species that have strong seasonal variations (Zhang et al., 2009). The MEIC emission inventory is verified to produce consistent aerosol precursor loadings with satellite observations (Li et al., 2010; Wang et al., 2010, 2012; Zhang et al., 2012; Liu et al., 2016). It has been widely used to study the trend of aerosol concentrations in China (Wang et al., 2013), the Asian air pollution outflow (Zhang et al., 2008; Chen et al., 2009), the

relative contribution of emission and meteorology to the aerosol variability (Xing et al., 2011), and the sensitivity of air
quality to precursor emissions (Liu et al., 2010).

The AR5 emission inventory is currently the default for CAM5. The method of mapping the MEIC emission inventory for CAM5 is described in the Supplement of this paper. In addition to the differences in the annual mean emissions in 2009 (Fig. S1), there are large differences in the seasonal variations of two emission inventories (Fig. 1). The AR5 anthropogenic emissions of $SO_2$, BC, and POM do not have seasonal variations. With the inclusion of emissions from biomass burning and
shipping for $SO_2$, BC, and POM, as well as volcanic source for $SO_2$, the total BC, POM, and $SO_2$ emissions have seasonal variations in the AR5 emission inventory. However, we should note that the seasonal variation in the AR5 emissions is rather weak since anthropogenic emission dominates in eastern China. This could be problematic since the severe winter haze events in northern China in recent years are often linked to the higher anthropogenic emission in winter. The MEIC emission is characterized by monthly variations for the emissions of $SO_2$, BC, and POM that peak in winter. The emission of
SOAG in both inventories shows a consistent seasonal variation that peaks in summer because the emissions of biogenic VOCs (isoprene and monoterpenes), which peak in summer, dominate the total SOAG emission.

We also carry out an additional CAM5 simulation using the decadal MEIC emission from 2002 to 2012 to examine the changes of ADREs due to emission. We choose these 11 years because China's economy recovered from a depression in 2002, and since then the $SO_2$ emission has started to grow dramatically and decreased after 2006 due to the application of
flue-gas desulfurization devices. After 2012 the annual emission rates did not change as dramatically as in the previous years.

For the first two simulations, we run CAM5 for year 2009 in a "constrained meteorology" mode where the model winds are nudged towards ERA-Interim (Dee et al., 2011) with 6 h relaxation timescale (Ma et al., 2013, 2014; Zhang et al., 2014). Climatological sea surface temperatures (SST) are prescribed in the two simulations. When simulating the decadal change
from 2002 to 2012, we use the reanalysis data in 2009 cyclically to nudge the model meteorological fields. The constrained meteorology technique facilitates the model-observation comparison of aerosols and gas species.Temperature and moisture are not nudged in this study. As evaluated in Zhang et al (2014), nudging temperature and moisture creates a large perturbation to the model state, resulting in unrealistic behaviour for cloud and convection parameterizations because these parameterizations are calibrated based on the free-running model climate. Because winds are constrained, the advection of
heat and moisture are constrained to some degree when the difference in local temperature and moisture between two simulations is small, but local source and sink terms for atmospheric temperature and moisture are computed according to the model fast processes (e.g., cloud processes) and land processes (due to prescribed SST). The changes in atmospheric temperature and moisture can in turn influence the gas- and aqueous-phase chemistry and aerosol loadings. The changes in aerosol loading will affect temperature through radiation. However, this local change in temperature is less than 1K (see
Section 8 in the Supplement).

We estimate the ADREs due to instantaneous impact of aerosol scattering and absorption on the Earth's energy budget. The ADREs are calculated by the difference between the "clear-sky" radiative flux in the standard model simulation and a diagnostic call to the model radiation code from the same simulation but neglecting the aerosol scattering and absorption. The horizontal resolution is $0.9^o \times 1.25^o$ and vertically there are 30 layers from surface to 2.25 hPa, with the lowest 4 layers

inside the boundary layer. We focus our analysis of model results over eastern China ($22^o$-$44^o$N, $100^o$-$124^o$E, the red rectangle in Fig. 2) where the strongest anthropogenic emissions are located.

## 2.2 Observational data

Satellite AOD retrievals from MODIS and MISR in 2009 are used to evaluate the model results. This study uses the monthly mean AOD from MODIS Terra collection 6 (MOD08_M3 product, https://ladsweb.nascom.nasa.gov/). We use the combined

AOD product from the Dark Target (Levy et al., 2010) and the Deep Blue (Hsu et al., 2004) algorithms. We also compare our simulations with ground-based AERONET AOD and single scattering albedo (SSA) retrievals at 12 sites in Mainland China, Hongkong, Taiwan, Japan, and Korea (shown in Fig. 2). Monthly averaged AOD and SSA in 2009 are calculated from the daily averages with the months that contain less than 3 daily values excluded.

Observation data of chemical compositions near the surface in China are collected from literatures (see Table S3 and the

references in the Supplement). The chemical compositions of particulate matter with diameter smaller than 2.5 μm ($PM_{2.5}$) are analyzed for sulfate, organic carbon (OC), BC, and SOA in these studies. The measured OC concentrations are multiplied by a factor of 1.4 for calculation of the total organic mass (i.e., POM) (Seinfeld and Pandis, 1998). Since the surface chemical composition data that covers a full year in 2009 is very limited, to compared at least one year's cycle of seasonal variation, we extend the range of time selection so that the observations are allow to continue from 2009 to 2010.

Many of the studies collected the samples continuously during April, July, October, and January to represent the concentrations in spring, summer, autumn, and winter. The observation in Xiamen was carried out for a full year of sample collection. Since we do not find the SOA measurements in 2009, we use the data in other years and are aware of the uncertainties due to the time difference. The geographical locations of these observations are shown in Fig. 2.

The ADREs have been estimated based on ground-based and satellite observations at different locations in China (Li Z. et

al., 2016). Table S4 in the Supplement lists the observations used in this study. Most of the data are from the Chinese Sun Hazemeter Network (CSHNET) (Xin et al., 2007; Li et al., 2010). The ADREs are consistently defined as difference of the irradiance at TOA, surface, and in the atmosphere with and without the presence of aerosols. The ADREs are either calculated by radiative transfer models using the measured or retrieved aerosol properties (AOD, SSA, phase function, Ångström exponent, and size distribution) and surface reflectance (Xia et al., 2007a; Li et al., 2010; Liu et al., 2011; Zhuang

et al., 2014), or derived from the fitting equation of irradiance measurements as a function of AOD (Xia et al., 2007b, c). Since the MEIC emission inventory is for anthropogenic aerosols, we only compare with observations at locations away from deserts that are less impacted by dust aerosol. For the same reason, the shortwave radiation is discussed since anthropogenic aerosols are mostly fine particles, the impact of which in the long-wave radiation can be ignored. All data

analyses are performed after cloud screening to ensure clear-sky conditions. Since the solar irradiance depends on solar
zenith angle (i.e., the time of the day), we compare with the measurements that are 24 h averaged. If both the TOA and the
surface ADREs are provided, we calculated the ADRE in the atmosphere by subtracting the ADRE at TOA by the ADRE at
surface.

## 3 Results

### 3.1 Impact of emission on the modeling of AOD, SSA, and surface concentration

*Aerosol optical depth (AOD)*

Figure 3 shows the spatial distributions of the annual averaged AOD over China simulated by CAM5-MAM3 using the
MEIC and the AR5 emissions in 2009 compared with satellite retrievals. Comparing with the spatial distribution of the
emissions (Fig. S1), the AOD distribution basically agree with the emission patterns of sulfate, BC, POM and dust aerosols,
which contribute about 85% of the total AOD. With the AR5 emission, the modeled AOD (0.19, including dust aerosol)
averaged over eastern China is 58.0% lower than the MODIS AOD (0.46) and 51.9% lower than the MISR AOD (0.40) (see
Table 1). The modeled AOD using the MEIC emission is 0.25, which is 30.4% higher than the AOD with the AR5 emission.
The impact of anthropogenic emissions on the modeled dust AOD is small (< 1.0% difference) due to slightly different
removal rates of dust resulting from the internal mixing with anthropogenic aerosols (e.g., sulfate). By using the MEIC
emission the AOD simulations is improved by 12.9% relative to MODIS and 14.7% relative to MISR compared with the
AR5 emission. This suggests that the emission uncertainty (bias) could account for 22.2%-28.4% of the underestimation of
AOD simulated by CAM5 with the AR5 emission in eastern China. Although the model bias is largely reduced by using
MEIC, the modeled AOD with the MEIC emission is still 45.1% lower than the MODIS AOD and 37.2% lower than the
MISR AOD. In spite of the underestimated magnitudes, both emission inventories reasonably reproduce the spatial
distribution of MODIS and MISR retrieved AOD (Fig. 3e and 3f). The Jing-Jin-Ji Region, Sichuan Basin, Shandong, Henan,
Anhui, Hunan, and Hubei Provinces are characterized by higher AODs than other parts of China, which is consistent with
the higher anthropogenic emissions in these regions.

In terms of the seasonal variation, the model simulates AOD maximums between 35$^o$N and 40$^o$N in early summer (from May
to July) with both emission inventories (Fig. 4), which is mostly due to dust aerosol transported from the west, while the
satellite retrievals do not show such strong dust emission and transport. The simulation with the MEIC emission captures
two observed AOD maximums in the spring (February to April) and in the autumn (August to October) around 30$^o$N, where
Sichuan Basin and central China are located, but the magnitudes are lower than the observations (Fig. 4). The simulation
using the AR5 emission fails to capture the first maximum and underestimate the second one even more than that with the
MEIC emission. By examining the model AOD components by species (Fig. S4), the first maximum is mostly due to sulfate
aerosol and to a less extent POM aerosol, and the second maximum is mostly due to sulfate. The satellite retrievals show a
third summer maximum in June, which is not captured by the model with both emission inventories. The time and location

of this observed AOD maximum complies with the $SO_2$ emission in MEIC (Fig. S3). Therefore, the observed maximum is probably due to efficient production of sulphuric acid gas ($H_2SO_4$) at higher temperatures and consequently formation of sulfate aerosol. Since the uncertainty of $SO_2$ emission is relative low ($\pm12\%$, Zhang et al., 2009) and the concentration of $SO_2$ is reasonably simulated by CAM5 (He et al., 2015), model underestimation cannot be explained by emission alone.

Other causes (e.g., wet scavenging, missing nitrate, particle size distribution, aerosol hygroscopic growth, etc.) in the model may be more responsible. For example, the model bias could be due to too much wet scavenging associated with the East Asian summer monsoon precipitation, which pushes too far to the north in summer compared with the Global Precipitation Climatology Project (GPCP) observations (Jiang et al., 2015). The CAM5-MAM3 does not include the treatment of nitrate aerosol, which can be an important aerosol component in East Asia (Gao et al., 2014).

More detailed comparisons with observations at 12 AERONET sites are given in Fig. 5. The model simulations using both emission inventories generally underestimate AOD compared with AERONET and satellite observations. The magnitudes of the AODs simulated with the MEIC emission are higher than that with the AR5 emission. The two simulations feature similar seasonal variations, for example, summer maximums at the sites north of 35 °N (Beijing, Xianghe, Xinglong, and SACOL). This is because the simulated AODs are dominated by sulphate and dust aerosols at these northern sites and by

sulphate aerosol at the southern sites (Taihu, Hongkong, and etc.) in both simulations (see Fig. S5 and Fig. S6 in the Supplement). Sulfate AOD peaks in summer in both simulations. In addition to the maximum in spring, dust AODs at the northern sites have two maximums in summer and autumn, which are suspicious and need further examination. The observed seasonality of AOD at northern sites features a maximum in July and a lower AOD in June, while the modeled AOD peaks in June that may due to overestimated dust aerosol. The model captures the seasonality of observed AOD in the

downwind regions but underestimates the magnitude of AOD by a factor of 2-3. AODs at the sites in 20-30°N (Taiwan and Hong Kong sites) are featured by the summer minimums in both observations and model results due to the scavenging of aerosols by the summer monsoon precipitation. The MEIC emission has a notable impact on AOD in Hong_Kong_PolyU site in all seasons and only has a small impact in winter at Taiwan sites (NCU_Taiwan, EPA_Taiwan, and Chen-Kung_Univ). The difference between the two emission inventories is not evident at Osaka and Shirahama.

The sensitivities of modeled AOD to the emission change between the two inventories are quite different for each aerosol species due to different aerosol refractive indexes (Table 1). 12.0% of POM emission difference results in 70.4% of the AOD difference. In contrast, 46.9% of the SOAG emission difference leads to only 17.4% of the AOD difference of SOA.

*Single scattering albedo (SSA)*

Figure 6 shows the modeled SSA using the MEIC and AR5 emissions and the comparison with the observations by

AERONET. The modeled SSA at Beijing, Xianghe, and Xinglong agrees with the AERONET data in terms of the strong seasonal variations of lower SSA in winter and higher SSA in summer, indicating higher fractions of absorbing aerosols in winter. However, the modeled SSA is systematically lower than the AERONET data. This indicates the significant

underestimation of light scattering aerosols (e.g., sulfate and POM). The SSA simulated with the MEIC emission is lower than that using the AR5 emission by up to 0.05 in winter, which is consistent with the higher BC emission in the MEIC emission. The SSA simulated with the MEIC emission in Taihu is slightly higher than that with the AR5 emission throughout the year, which is consistent with the higher MEIC emission of sulfate. Outside Mainland China the modeled SSA agrees with AERONET data reasonably well at the Hong Kong, Taiwan and Japanese sites, although underestimations can be found in some months.

*Aerosol surface concentrations*

Figure 7 compares the modeled surface concentrations of sulfate, BC, POM, and SOA with the observations of chemical compositions. The surface concentrations of these aerosol species are generally underestimated in the model with both emission inventories, which is consistent with the underestimations of AODs. The concentrations of sulfate aerosol are underestimated by about a factor of 3 (the linear regression slope of 0.35) using the MEIC emission but is improved compared with about a factor of 5 (the linear regression slope of 0.18) using the AR5 emission. Since the concentration of $SO_2$ is reasonably well simulated by CAM5 over east Asia (He et al., 2015), there could be a fundamental error in the model treatment of the conversion from precursor gases to secondary aerosols. The POM and BC surface concentrations are significantly improved by the MEIC emission due to higher emission rates especially in winter. The root mean square errors (RMSEs) using the MEIC emission are 10.01, 14.63, 3.32, 6.58 µg m$^{-3}$ for sulfate, POM, BC, and SOA, respectively, which are smaller than RMSEs of 13.38, 19.21, 3.97, 8.38 µg m$^{-3}$ using the AR5 emission. The coefficients of determination ($R^2$) between model simulations and observations of all these species are also improved. Considering that most observations are carried out at single points and at altitudes close to the surface, the underestimation could be partly due to the coarse model horizontal and vertical resolutions. The model with a coarse horizontal resolution does not account for the subgrid variability of aerosols (Qian et al., 2010). With the coarse vertical resolution, aerosol species are assumed to be well mixed in the bottom model layer with a thickness of about 60 m, which may lead to low biases compared with the observations.

**3.2 Distinct impact of emission and atmospheric processes on aerosol seasonal variations**

Observed surface concentrations at 10 locations in China show that the primary and secondary aerosols have distinct seasonal variations (Fig. 8). The observed surface concentrations of primary aerosols (BC and POM) at all locations show maximums in winter, suggesting that their seasonal variations are mainly controlled by the emission. The MEIC emission significantly improves the modeled seasonal variations compared with the AR5 emission that has no seasonal variations of POM and BC emissions. In contrast, the observed concentrations of sulfate in northern China (Chengde, Shangdianzi, Beijing, Tianjin, Shijiazhuang, Zhengzhou) are characterized by summer maximums. This is due to a higher photochemical production rate in summer (Wen et al., 2015). The modeled concentrations of sulfate also show their maximum in summer. This feature is commonly seen for many climate models. The concentrations of sulfate in the southern cities (Xiamen and Guangzhou) do not have summer maximums due to the Asian summer monsoon with strong winds and precipitation.

We examine the processes that determine the concentrations of sulfate in the model, including gas-phase and aqueous-phase production, dry and wet scavenging, as well as the controlling meteorological variables (Fig. 9). The MEIC emission of $SO_2$ peaks in winter in northern China due to heating in the domestic section, whereas the AR5 emission does not have seasonal variations (Fig. S7). Obviously, the surface concentrations of sulfate aerosol cannot be explained by emission alone and the atmospheric processes are more likely responsible for the seasonality. We find that the simulated seasonal variations of

surface concentrations of sulfate aerosol are controlled by the gas-phase and aqueous-phase production processes and to a less extent by the emission of $SO_2$. The gas-phase chemistry is most active in summer due to the temperature-dependence of the oxidation rate of $SO_2$ by OH. Also the oxidation rate depends on the concentration of OH radical, which is highest due to efficient photochemical reactions in summer. The aqueous-phase formation of sulfate aerosol also peaks in summer due to higher relative humidity and thus more cloud water. Although the MEIC $SO_2$ emissions peak in winter, both the gas-phase

and aqueous-phase oxidations are less efficient in winter, which results in lower concentrations of sulfate aerosol than in summer.

We notice that some other observations show different seasonality of sulfate aerosol from the model results. For example, observations from CAWNET (Zhang et al., 2012) show that concentrations of sulfate aerosol in the northern Chinese cities (e.g., Gucheng and Zhengzhou in Fig. S8) peak in winter as opposed to summer in spite of a minor maximum in summer.

The observed seasonal variations at two pairs of nearby sites from CAWNET and our study (Gucheng 2006-2007 versus Beijing 2009-2010, Zhengzhou 2006-2007 versus 2009-2010) are different from each other. This may reflect that the relative contributions of the emissions and the atmospheric processes in determining the concentration of sulfate change with years and locations. It is also possible that some mechanisms of sulfate aerosol formation for these CAWNET sites, which are especially important in winter, are not properly modeled or missing in the model. For example, the aqueous-phase oxidation

of $SO_2$ in pre-existing aerosols is not modeled, which is important in explaining the winter haze in China (Wang et al., 2016; Cheng et al., 2016).

Having the same "constrained" meteorology for the two simulations with different emission inventories provides us with an opportunity to examine the impact of emission versus atmospheric processes on the seasonality of aerosols. The longitudinal averaged BC burden in the simulation with the MEIC emission shows a strong seasonal variation between 25 and 40 $^{o}$N with

higher burden in winter (Fig. 10a), which corresponds with the seasonal variation of BC emission (Fig. 10b). Since there is no seasonal variation for BC aerosol in the AR5 emission (Fig. 10d), the seasonal variation of BC concentrations can only be due to the impact of atmospheric processes in the AR5 emission run (Fig. 10c). The winter peak is also seen for the AR5 run most likely due to stagnant wind fields for dispersion in winter. The summer minimums are due to wet scavenging by the monsoon precipitation. Fig. 10 indicates that seasonal variations of both the emission and atmospheric processes play

important roles in determining the seasonal variation of BC concentrations.

The distinct impacts of emissions and atmospheric processes that are associated with meteorological factors on the seasonal variations of primary (e.g., BC) and secondary aerosols (e.g., sulfate) are further demonstrated in Figure 11. The seasonal variation of differences in the longitudinally averaged burden of BC between the two emission runs resembles closely the

pattern of differences in the emission of BC (Fig. 11a and Fig. 11b). However, the dependence of seasonal variation of burden of sulfate on the emission of $SO_2$ is less evident (Fig. 11c and Fig. 11d). The difference of $SO_2$ emission between the two inventories in 30-45°N is amplified by the production and condensation of $H_2SO_4$ gas that are favored at higher temperatures in summer. This larger difference of sulfate burden between the two emission runs is obviously aligned with higher temperatures between 30°N and 45°N in summer (May to July) (Fig. 11e). In contrast, although there is a comparable difference in the $SO_2$ emission between 35°N and 40°N in cold seasons (November to March), the difference of sulfate burden is not as evident due to the fact that low temperatures inhibit the production of sulfate. Wet scavenging by clouds and precipitation helps to reduce the concentrations and their absolute differences in southern China during spring and summer (Fig. 11f and Fig. 11g). Higher wind speeds north of 35°N in winter (Fig. 11h) for aerosol dispersion help to explain the small difference of sulfate burden in spite of the evident difference of $SO_2$ emission there. Stagnant wind field that propagates from 22°N to 35°N in spring and from 35°N to 22°N in autumn makes the impact of difference in emissions on the large differences of BC and sulfate burdens between the two simulations prominent in the corresponding seasons and regions. Due to the complex atmospheric processes, the spatiotemporal patterns of secondary aerosol burdens follow less closely to their precursor gas emissions compared with primary aerosols.

### 3.3 Impact of aerosol-meteorology interactions

Changes in the aerosol radiative forcing will alter atmospheric temperature and moisture in the model, and can, in turn, influence gas- and aqueous-phase chemistry, the boundary layer stability, and eventually the aerosol loadings. In this study, the differences in meteorological fields between the two simulations reflect aerosol effects through fast processes (e.g., cloud process) and land processes since the horizontal winds and climatological SST are prescribed. We examine the differences in the temperature and relative humidity in our simulations with the two different emission inventories. The changes in temperature and relative humidity is less than 1 K and 3%, respectively (Figure S10). These are small changes compared to the seasonal variations and therefore may not affect our findings on the impacts of emissions and atmospheric processes on aerosol loadings. More discussion on the effect of aerosol-meteorological interactions is provided in Section 8 of the Supplement. Nevertheless, aerosol-meteorology interactions could be significant through the interplay between aerosol and boundary layer processes (Wang et al., 2013) as well as cloud microphysics (Wang et al., 2011). Specifically, the effects of enhanced light absorption by aerosols (e.g., BC) as well as the aging of BC may change the boundary layer stability (Khalizov et al., 2009; Peng et al., 2016). We suggest assessing these feedbacks by using a fully coupled, free-running earth system model with more detailed treatment of BC aging, boundary layer, and cloud microphysics in the future.

### 3.4 Impact of emission on the modeling of ADREs

Figure 12 shows the spatial distribution of annual averaged shortwave ADREs in China simulated using the MEIC and the AR5 emissions due to all aerosol species at TOA, surface, and in the atmosphere. The TOA radiative cooling effect is evident in eastern China due to anthropogenic aerosols. At some parts of the southwestern China the ADRE at TOA is positive due to strong BC absorption in the atmosphere. The most pronounced surface cooling and atmospheric warming are

located in the northern China and the Sichuan Basin, which is consistent with the spatial patterns of the emissions. In these locations the surface and atmospheric differences of the ADREs between the two simulations are also significant.

As shown in Table 2 the annual averaged cooling effect at TOA is reduced (more negative) by -0.91 W m$^{-2}$ (22.3%) by all aerosols using the MEIC emission (-5.02 W m$^{-2}$) compared with that using the AR5 emission (-4.11 W m$^{-2}$). At the surface there is a strong cooling effect of -18.47 W m$^{-2}$ using the MEIC emission, which is reduced (more negative) by -3.48 W m$^{-2}$ (23.3%) compared with that using the AR5 emission (-14.99 W m$^{-2}$). The atmospheric warming effect of all aerosols using the MEIC emission is estimated to be 13.45 W m$^{-2}$, which is 2.57 W m$^{-2}$ (23.6%) stronger than the estimation made by the AR5 emission (10.88 W m$^{-2}$) over eastern China.

Table 2 also shows the annually averaged ADREs over eastern China by individual aerosol species. The ADREs of SOA are not shown due to its large emission uncertainty. Due to larger AODs simulated with the MEIC emission, the ADREs by each aerosol species are larger than the ADREs using the AR5 emission by 33.6% to 47.2% at TOA. Tables 1 and 2 show that over eastern China 12.0% to 46.9% difference of the anthropogenic emission rates of various aerosol species results in 30.4% difference of the total AOD of all species (including anthropogenic and natural aerosols) and 22.3%, 23.3%, and 23.6% differences of the ADREs at TOA, the surface, and in the atmosphere, respectively. The impacts of the emission on AOD and ADREs are significant.

The normalized radiative effect (NRE) represents the radiative effect efficiency per unit aerosol optical depth (Schulz et al., 2006). The light scattering aerosols (sulfate and POM) have very similar negative NREs (-31.77 and -33.84 W m$^{-2}$ $\tau_{aer}^{-1}$ with the MEIC emission hereafter). The light absorbing BC aerosol shows a much higher positive NRE (100.52 W m$^{-2}$ $\tau_{aer}^{-1}$) which is comparable to the mean NREs of the AeroCom models (153 W m$^{-2}$ $\tau_{aer}^{-1}$) considering the wide range of the estimates among the models (28 to 270 W m$^{-2}$ $\tau_{aer}^{-1}$) (Schulz et al., 2006). The NREs of BC are much higher than the other aerosol species, especially the warming in the atmosphere (305.50 Wm$^{-2}$ $\tau_{aer}^{-1}$). This indicates that the ADREs are much more sensitive to BC aerosol burden than the other aerosol species and highlights the importance of the BC emission and concentration to correctly represent the ADREs in the model. BC also makes the largest contributor to the ADRE in the atmosphere and at the surface. We note that the ADREs of light scattering aerosols (sulfate and POM) in the atmosphere are also warming effects. The explanation is that coating of these scattering aerosols on BC increases the absorption capability of the internally mixed aerosol particles (i.e., particles in the same aerosol mode with BC) (Chung et al., 2011).

The modeled spatial distributions of ADREs of BC in summer and winter at the surface and in the atmosphere are shown in Figs. 13 and 14, respectively. With the AR5 emission, the averaged ADREs over eastern China in winter (-4.40 W m$^{-2}$ at surface and 6.11 W m$^{-2}$ in the atmosphere) are close to the ADREs in summer (-4.40 W m$^{-2}$ at surface and 6.28 W m$^{-2}$ in the atmosphere). Due to the higher MEIC BC emission in winter, the cooling effect of BC at the surface is much more significant using the MEIC emission (-7.35 W m$^{-2}$) than the AR5 emission averaged over eastern China (Fig. 13). Likewise the warming effect of BC in the atmosphere with the MEIC emission (10.50 W m$^{-2}$) is nearly twice as much as that using the AR5 emission (Fig. 14). Driven by the same constrained meteorology, the MEIC emission results in much stronger seasonal variation of ADREs of BC than the AR5 emission.

Figure 15 shows the comparison between the measured and modeled ADREs at TOA, surface, and in the atmosphere over China. Observations from 25 nationwide stations shows that clear-sky ADREs are characterized by a strong radiative heating in the atmosphere, which implies a substantial warming in the atmosphere and cooling at the surface (Li et al., 2007; Li et al., 2010). Model simulations show small ADREs ($\sim$ -10 to -2 W m$^{-2}$) at TOA with both the MEIC and AR5 emission inventories, while the measurements gives a larger range of ADREs at TOA ($\sim$ -14 to 2 W m$^{-2}$). At the surface and in the atmosphere, the modeled ADREs using the MEIC emission inventory at most locations are within a factor of 2 compared with observations. The MEIC emission inventory produces better agreement with the observations than the AR5 emission inventory.

## 4. Decadal trend of ADRE

The uncertainty of aerosol emissions used in climate models could affect the historical and future aerosol effects simulated by the models. Here in this section, we assess the changes in ADREs as the change in the emission in the past decade and compare them with the difference that results from the use of the two emission inventories.

The magnitude and structure of aerosol and precursor gas emissions in China have significantly changed during the last decade (Zhao B. et al., 2013; Lu et al., 2011; Kang et al., 2016). The emission trend used in this study is estimated by the MEIC development team based on their knowledge on the evolution of activity level and technology in China (see Fig. S9 in the Supplement). Figure 16 shows that the decadal trend of ADRE agrees with the trend of emissions (Fig. S9). The warming in the atmosphere and the cooling at the surface were both enhanced with the increase of emissions of $SO_2$, BC, and POM from 2003 to 2006. The ADRE at TOA only decreased slightly indicating more energy lost from the atmosphere-earth system. From 2006 to 2009, the changes of ADREs were not significant due to the stabilized emission of BC. Since 2010, the warming in the atmosphere and the cooling at the surface both increased due to the increased emission of $SO_2$ and BC. The changes of ADREs at surface and in the atmosphere from 2002 to 2003 may reflect the complicated interactions between sulfate and BC/POM in eastern China, enhancing the BC/POM wet scavenging due to sulfate coating.

The ranges of the decadal changes of ADREs at TOA (-0.45 to 0.07 W m$^{-2}$), at surface (-0.99 to 0.19 W m$^{-2}$), and in the atmosphere (-0.20 to 0.60 W m$^{-2}$) are smaller than the differences of ADREs between MEIC and AR5 emissions in 2009 (-0.91 W m$^{-2}$, -3.48 W m$^{-2}$ and 2.57 W m$^{-2}$). It highlights the uncertainty of the emission inventories and the need of constraining the emission inventories of aerosols and precursor gases by in-situ and satellite observations.

## 5 Summary and Conclusions

Anthropogenic aerosols in East Asia have substantial effects on regional air quality and climate. However, global climate models generally have low biases in anthropogenic aerosol burdens in this region (Shindell et al., 2013), and thus the aerosol radiative effects may be underestimated. The reasons behind the low biases are unclear, but may include the bias in aerosol emissions, missing of some aerosol processes, coarse model resolutions, etc. In this study, we simulated the aerosol concentrations, optical depth, and the radiative effects in eastern China using the Community Atmospheric Model version 5

with the 3-mode Model Aerosol Module (CAM5-MAM3). A technology-based emission inventory, Multi-resolution Emission Inventory for China (MEIC), was implemented into CAM5-MAM3 and results were compared with the simulation using the default IPCC AR5 emission inventory.

We found that the MEIC emission improves the annual mean AOD simulations in eastern China by 12.9% compared with the MODIS observations and 14.7% compared with the MISR observations, which explains 22.2%-28.4% of the AOD underestimation simulated with the AR5 emission. The MEIC emission generally reproduces the AOD spatial distribution although AOD is still underestimated compared with the MODIS and MISR satellite retrievals.

CAM5 with the MEIC emission captures the AOD maximums around $30^{o}$N in spring and autumn better than CAM5 with the AR5 emission. However, both emission runs underestimate the AOD maximum around $30^{o}$N in summer, which coincides with the modeled summer monsoon precipitation that pushes too far to the north. Wet scavenging by summer monsoon precipitation should be reasonably represented since it significantly affects the model AODs. The modeling of dust aerosol is also of particular importance in northern China.

The simulated surface concentrations of both primary and secondary aerosols are improved by using the MEIC emission compared with that modeled by the AR5 emission. The MEIC emission leads to better agreement with the observed seasonal variations of the primary aerosols (i.e., POM and BC) than the AR5 emission in term of seasonal variation, but the concentrations are still underestimated. This implies that the atmospheric loadings of primary aerosols are closely related to the emission, which may still be underestimated over eastern China. In contrast, the seasonal variations of secondary aerosols (i.e., sulfate) depend more on the aerosol processes (e.g., gas and aqueous phase chemistry) associated with the meteorological factors (e.g., temperature, relative humidity, winds) and to a less extent on the emission. Analysis of the aerosol processes in the model shows the gas phase and in-cloud aqueous-phase formation of sulfate aerosol peaks in summer due to higher temperature, photolysis rate, and relative humidity. Therefore, it is suggested that the emissions of secondary aerosols alone cannot explain all the low biases in the model over eastern China. Aerosol processes in CAM5 should be revisited. For example, we notice that some other observations (e.g., CAWNET) show winter peaks of the sulfate concentration in northern China in different years and locations, which may reflect that some mechanisms, such as production through heterogeneous reactions of $SO_2$ on pre-existing aerosols, are important for these observation sites and should be included in the model. Observations and regional air quality modeling with more complex chemistry reveals the importance of sulfate production on mineral dust through gas-phase uptake or heterogeneous reactions in increasing the $PM_{2.5}$ concentrations and the mass fractions of secondary inorganic aerosols (Wang Y.X. et al. 2014; Huang et al., 2014; Zheng et al., 2015; Dong et al., 2016). The coexistence of $NO_2$ and $SO_2$ promotes heterogeneous production of sulfate aerosol under high relative humidity (He et al., 2014; Chen et al., 2016; Wang Y.S. et al., 2014). The aqueous-phase oxidation of $SO_2$ by $NO_2$ (Wang et al., 2016; Cheng et al., 2016) or $O_3$ (Palout et al., 2016) is efficient to form sulfate aerosol under high relative humidity and $NH_3$ neutralization conditions. Nitrate aerosol is not modeled in CAM5 and could be an important contributor to AOD in eastern China. It is also possible that the default accommodation coefficient of $H_2SO_4$ gas is set too high in CAM5-MAM3, which results in too efficient condensation and less efficient nucleation of $H_2SO_4$ to form

sulfate aerosol (He and Zhang, 2014). The aerosol-meteorology interactions do not influence our findings on the relative importance of emission and atmospheric processes since model winds and SST are prescribed and thus the differences of temperature and relative humidity are small. However, we suggest further investigation into the impact of aerosol-meteorology interaction on the aerosol loading using a fully coupled, free-running earth system model with more detailed treatment of BC aging, boundary layer, and cloud microphysics in the future.

Different emissions have substantial effects on the aerosol direct radiative effects (ADREs). By using the MEIC emission, the annual averaged ADREs at TOA and at the surface over eastern China are reduced (more negative) by -0.91 W m$^{-2}$ and -3.48 W m$^{-2}$, respectively, while the warming in the atmosphere is increased by 2.57 W m$^{-2}$. The ADREs between the MEIC and AR5 emissions with all aerosol species (including natural dust) are increased by 22.3%, 23.3%, and 23.6% at TOA, surface and in the atmosphere, respectively. The ADRE is more sensitive to BC aerosol burden than the other aerosol species. Due to the higher MEIC BC emission in winter, the warming effect of BC in the atmosphere and the cooling effect at the surface are much higher than those using the AR5 emission. This implies that enhanced BC loading in winter will lead to strong atmospheric inversion (Wang et al., 2015; Ding et al., 2016). In summary, 12.0% to 46.9% difference of the emission rates of different aerosol species results in 30.4% difference of the total AOD, and about 22% difference of the ADREs averaged over eastern China. The impacts of the emission on AOD and ADREs are significant.

By examining the change of ADRE from 2002 to 2012 using the estimation of emissions made by the MEIC development team, we find that the decadal changes of ADREs are smaller than the differences of ADREs simulated by the two emission inventories at TOA, surface and in the atmosphere over eastern China. This indicates that there is an urgent need to constrain the emission inventories of aerosols and precursor gases by in-situ and satellite observations.

This research highlights the critical importance of improving emissions of aerosols and precursor gases as well as the aerosol processes for the modeling of aerosols and aerosol radiative effects in eastern China, although any improvement in our understanding of the underlying processes would be equally valuable anywhere else. We note that modeled AOD and surface concentrations are still underestimated in CAM5 even with the MEIC emission. Yet, if the estimations of MEIC emissions in trace gases do not suffer similar biases as in the AOD, our findings would help affirm a fundamental error in the conversion from precursor gases to secondary aerosols as hinted in other recent studies following different approaches. Recently, the Community Emission Data System (CEDS) is newly released and is intended for use in CMIP6 (Hoesly et al., 2017). The CEDS emission for eastern China is comparable with MEIC (see Section 3 in the Supplement) since CEDS is scaled to country-level inventories, i.e., MEIC for China (Li et al., 2017). Without improvements in the aerosol process, the similar low-bias over eastern China in CMIP5 GCMs are expected in CMIP6. There also exist aspects other than aerosol process that potentially leads to the low bias. The CAM5 model with a horizontal resolution of 0.9$^{o}$×1.25$^{o}$ may miss the subgrid aerosol variability (Qian et al., 2010) as well as not able to capture the collocation between aerosols and clouds important for aerosol wet scavenging (Ma et al., 2014). CAM5-MAM3 may also miss some important aerosol species (e.g., nitrate) which can have similar mass burdens as sulfate in eastern China (Gao et al., 2014). Current work is under the way to increase the

500 model resolution and to implement nitrate aerosol in CAM5-MAM3. The impacts of these new developments on aerosols in East Asia will then be re-evaluated.

In this study, as the first step the impacts of a new emission inventory on the simulations of AOD, aerosol concentrations and ADREs in east China are examined. Future studies of impacts on clouds, precipitation and atmospheric circulation in east China and anywhere else will be conducted. Using a global climate model with interactions between aerosols, cloud, 505 precipitation and meteorology, we will be able to study the potential impacts of climate changes on pollution conditions in China. A predominant climatic phenomenon in China is East Asian monsoon, and thus the impacts of monsoon variability on air pollution have gained a lot of attentions (Wu et al., 2016). Long-term (~30 years) simulation will be needed to study the impact of climate changes aerosol in China.

*Acknowledgements.* The authors would like to acknowledge the use of computational resources (ark:/85065/d7wd3xhc) at the NCAR-Wyoming Supercomputing Center provided by the National Science Foundation and the State of Wyoming, and supported by NCAR's Computational and Information Systems Laboratory. This work was supported by National Natural Science Foundation of China (Grant no. 41705125) and by the Ministry of Science and Technology of China (Grant no. 2013CB955804). Both T. Fan and C. Zhao were supported by the Fundamental Research Funds for the Central Universities 515 (Grant No. 310400090). Po-Lun Ma acknowledges internal support from Pacific Northwest National Laboratory, which is operated for the Department of Energy by Battelle Memorial Institute under contract DE-AC05-76RL01830.We thank the AERONET PI investigators and their staff for establishing and maintaining the 12 sites used in this investigation.

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

**Table 1. AOD averaged over eastern China in 2009 simulated using the MEIC and the AR5 emissions.**

| Species | MEIC AOD | AR5 AOD | (MEIC-AR5)/AR5 AOD | (MEIC-AR5)/AR5 Emission |
|---|---|---|---|---|
| Sulfate | 0.085 | 0.059 | 44.3% | 12.6% |
| BC | 0.030 | 0.021 | 42.6% | 13.4% |
| POM | 0.044 | 0.026 | 70.4% | 12.0% |
| SOA | 0.031 | 0.026 | 17.4% | 46.9% |
| Dust | 0.057 | 0.056 | 1.0% | 0.0% |
| Sea salt | 0.006 | 0.005 | 4.2% | 0.0% |
| All aerosols | 0.252 | 0.193 | 30.4% | - |

**Table 2. Aerosol direct radiative effects (ADRE) and the normalized radiative effect (NRE) averaged over eastern China in 2009 simulated using the MEIC and the AR5 emissions.**

|  | Species | MEIC ADRE, Wm$^{-2}$ | AR5 ADRE, Wm$^{-2}$ | (MEIC-AR5)/AR5 ADRE, % | MEIC NRE, Wm$^{-2}$ $\tau_{aer}^{-1}$ | AR5 NRE, Wm$^{-2}$ $\tau_{aer}^{-1}$ | *Schulz et al.,* [2006] NRE, Wm$^{-2}$ $\tau_{aer}^{-1}$ |
|---|---|---|---|---|---|---|---|
| TOA | All aerosols | -5.02 | -4.11 | 22.3% | -20.83 | -22.05 | |
| | Sulfate | -2.62 | -1.96 | 33.6% | -31.77 | -33.91 | -19 |
| | | | | | | | (-32 to -10) |
| | BC | 2.51 | 1.81 | 39.1% | 100.52 | 99.64 | 153 |
| | | | | | | | (28 to 270) |
| | POM | -1.38 | -0.94 | 47.2% | -33.84 | -36.70 | -19 |
| | | | | | | | (-38 to -5) |
| Surface | All aerosols | -18.47 | -14.99 | 23.3% | -72.5 | -76.06 | |
| | Sulfate | -3.40 | -2.58 | 31.7% | -40.36 | -43.78 | |
| | BC | -5.73 | -4.40 | 30.4% | -204.98 | -211.71 | |
| | POM | -2.72 | -1.78 | 52.5% | -63.73 | -68.04 | |
| Atmosphere | All aerosols | 13.45 | 10.88 | 23.6% | 51.67 | 54.01 | |
| | Sulfate | 0.79 | 0.62 | 26.0% | 8.58 | 9.87 | |
| | BC | 8.25 | 6.21 | 32.9% | 305.50 | 311.35 | |
| | POM | 1.33 | 0.84 | 58.4% | 29.89 | 31.35 | |

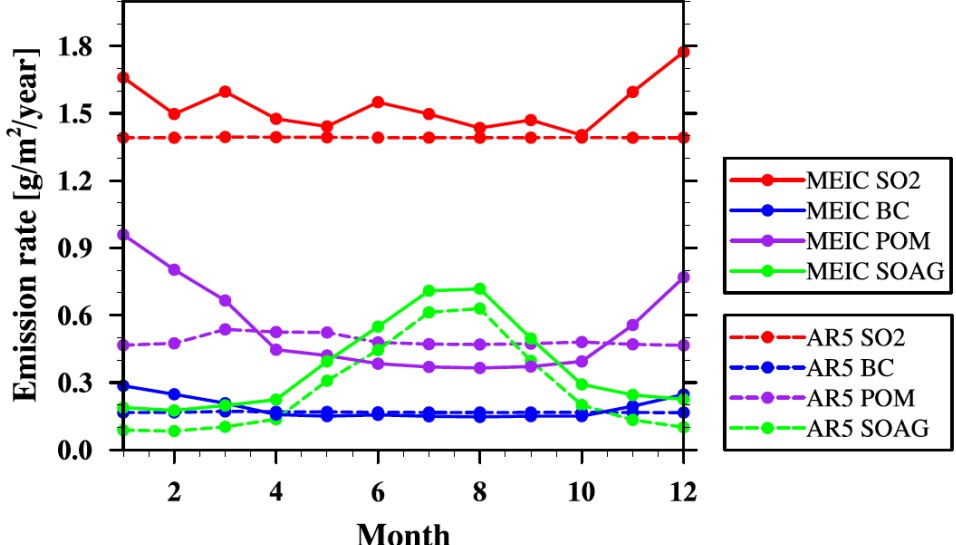


**Figure 1. Seasonal variations of sulfate, BC, POM, and SOAG in the MEIC emission and the AR5 emission in China for year 2009.**


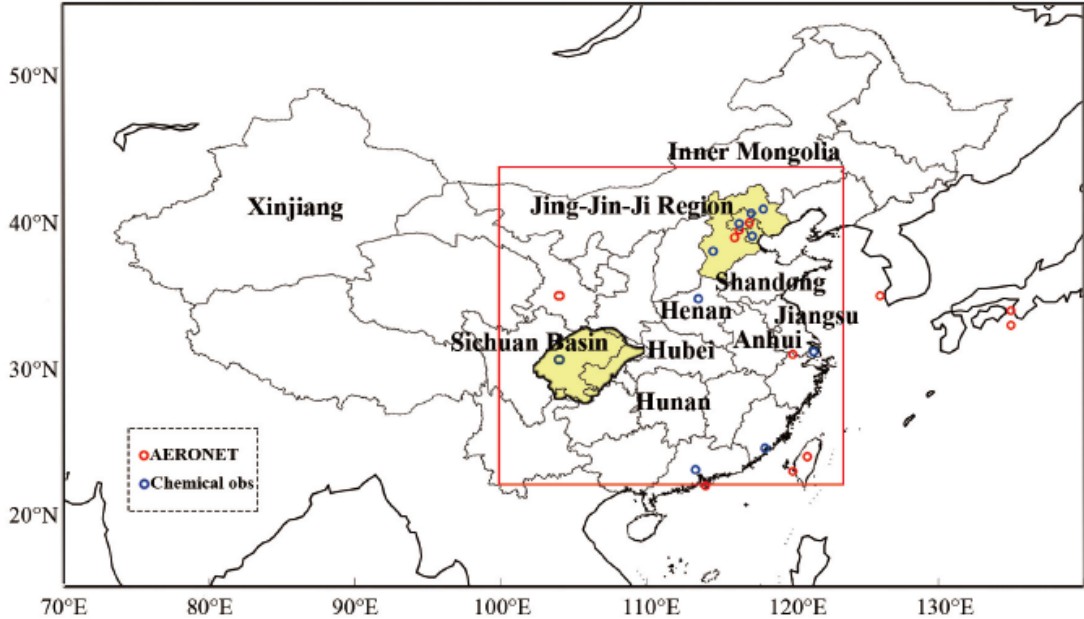

**Figure 2. Geographical locations of the AERONET sites and chemical composition sites where the observational data are used in this study. The provinces and regions mentioned in the context are marked. The red rectangle denotes eastern China (22-44$^o$ N, 100-124$^o$ E).**


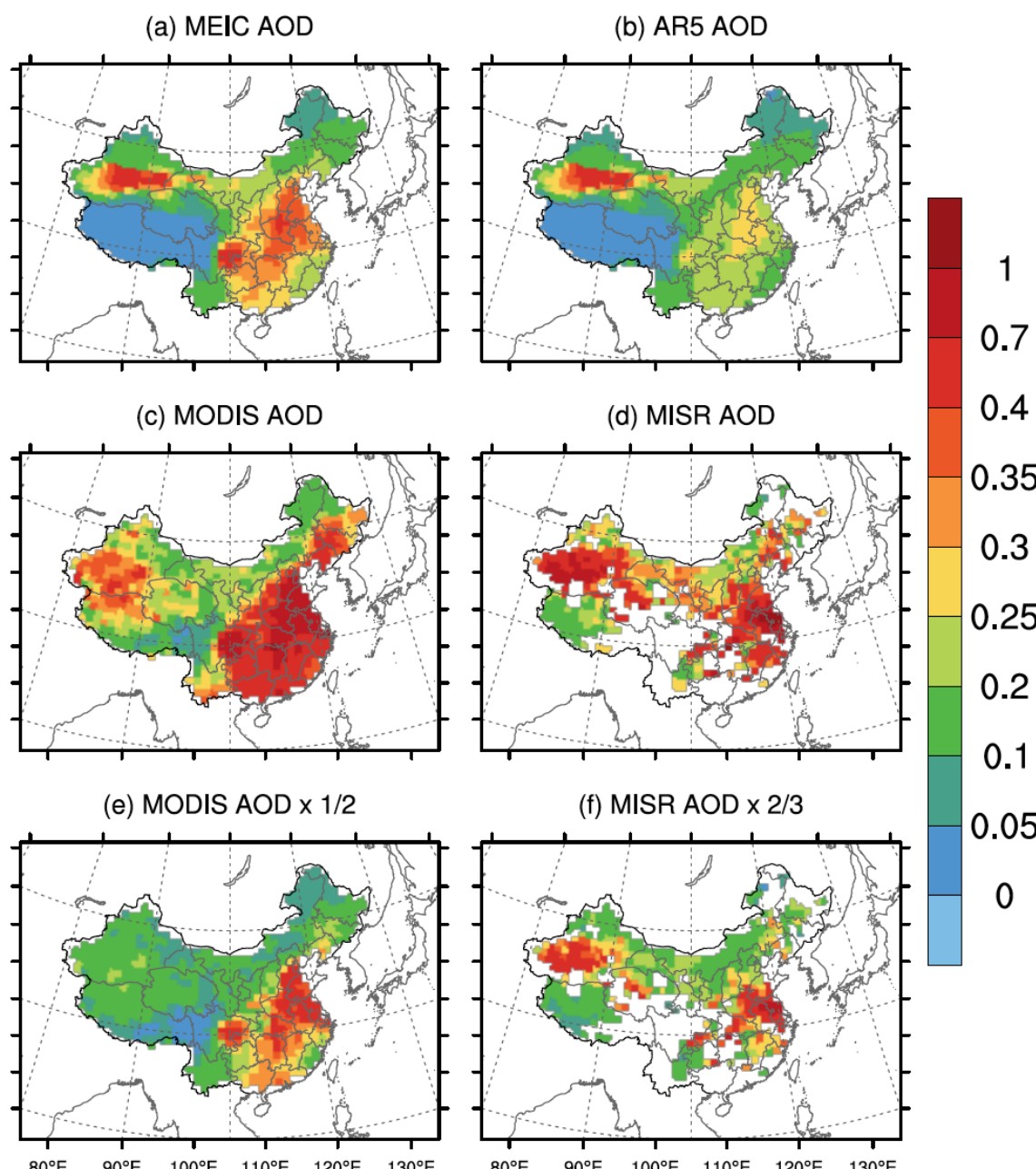

**Figure 3. Spatial distributions of annual averaged AOD at 550 nm over China in 2009 simulated by CAM5-MAM3 using (a) the MEIC emission, (b) the AR5 emission, observed by (c) MODIS and (d) MISR satellites, (e) MODIS AOD scaled by one half, and (f) MISR AOD scaled by two thirds. The scaling factors are approximately the ratios between the modeled AOD with the MEIC emission and retrieved AODs averaged over eastern China.**

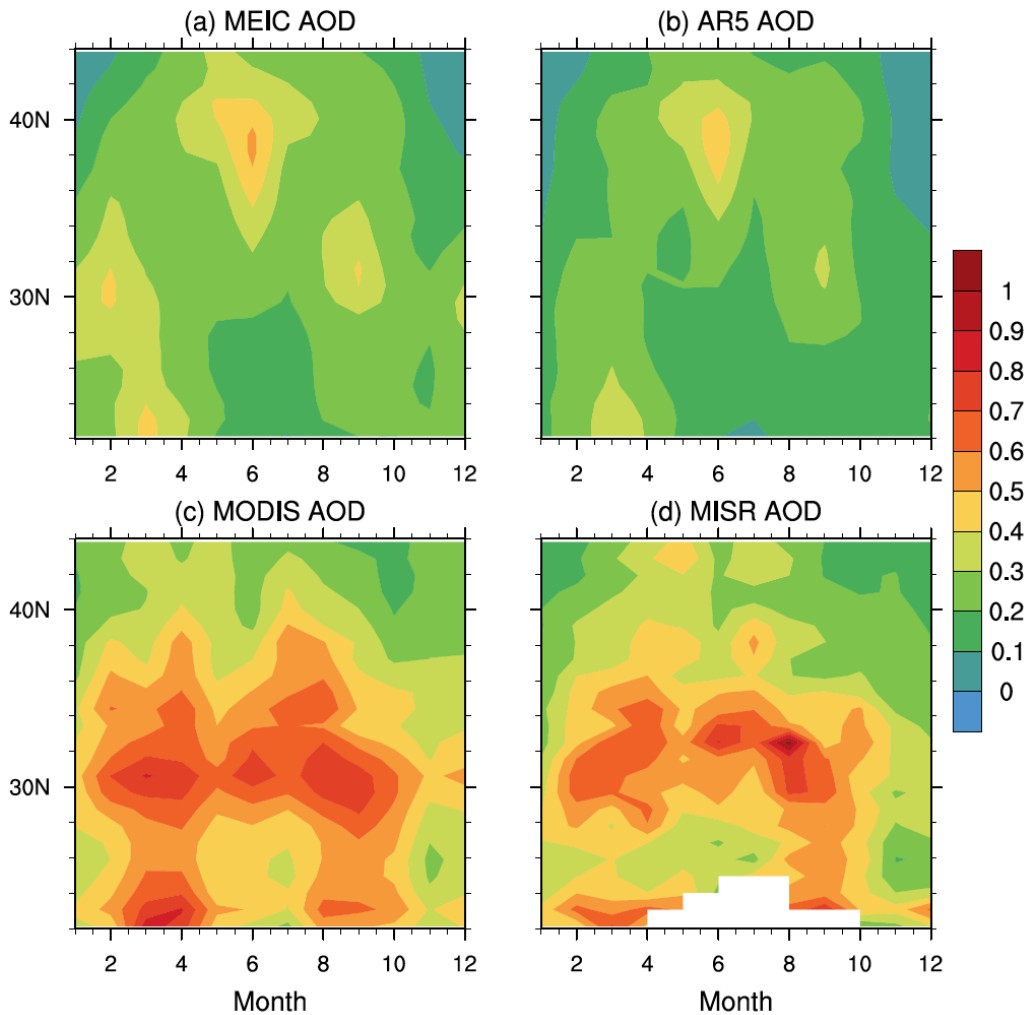

**Figure 4.** The seasonal variation of longitudinal averaged ($100^o$ E-$124^o$ E) AOD at 550 nm over eastern China simulated by CAM5-MAM3 using (a) the MEIC emission, (b) the AR5 emission, observed by (c) MODIS, and (d) MISR satellites in 2009.

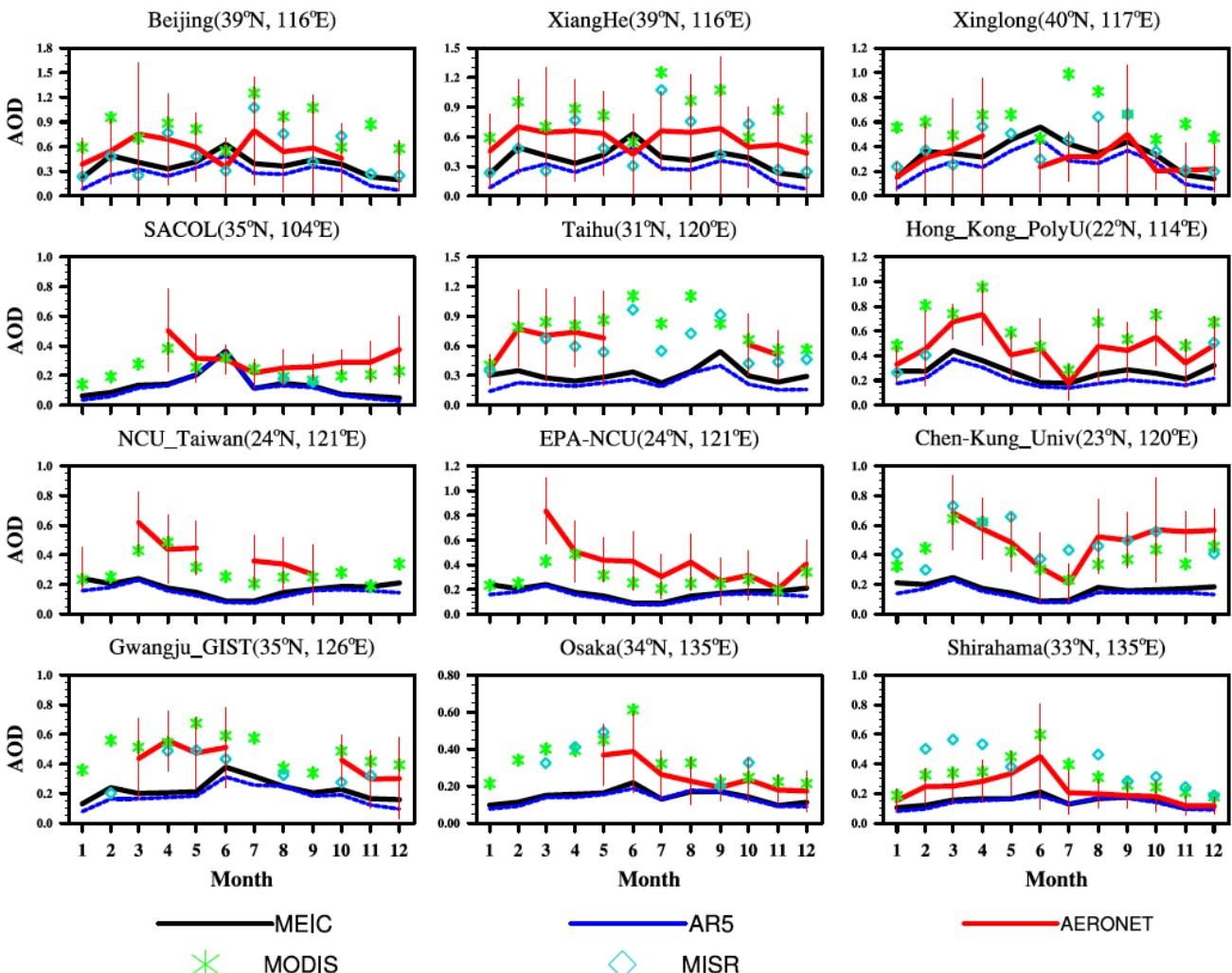

**Figure 5. Monthly averaged AOD simulated by CAM5-MAM3 using the MEIC emission and the AR5 emission compared with the AERONET, MODIS and MISR observations at 12 AERONET sites in and around China. The error bars represent one standard deviation of the daily AERONET observations within the month.**

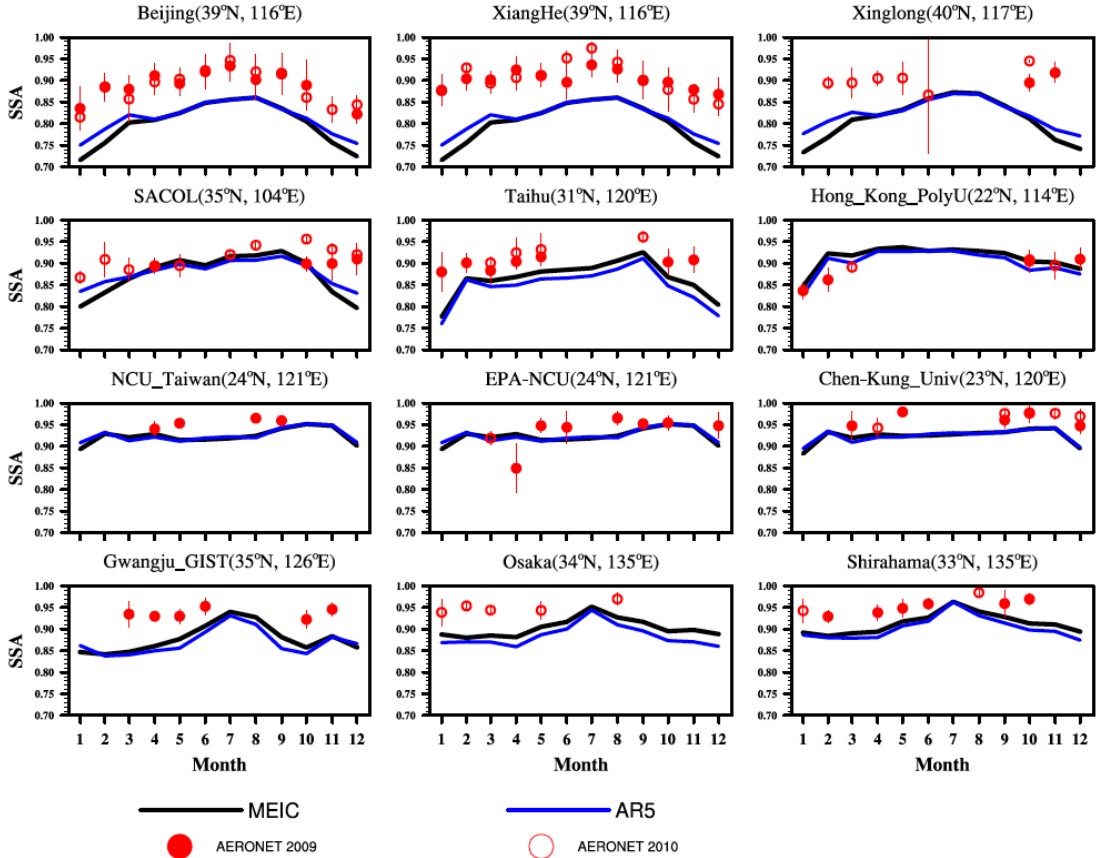

**Figure 6. The seasonal variation of SSAs simulated by CAM5-MAM3 using the MEIC emission and the AR5 emission for year 2009 and observed by AERONET (red solid circles for year 2009 and hollow circles for year 2010) at 12 AERONET sites in and around China. Error bars stand for one standard deviations of the observations.**

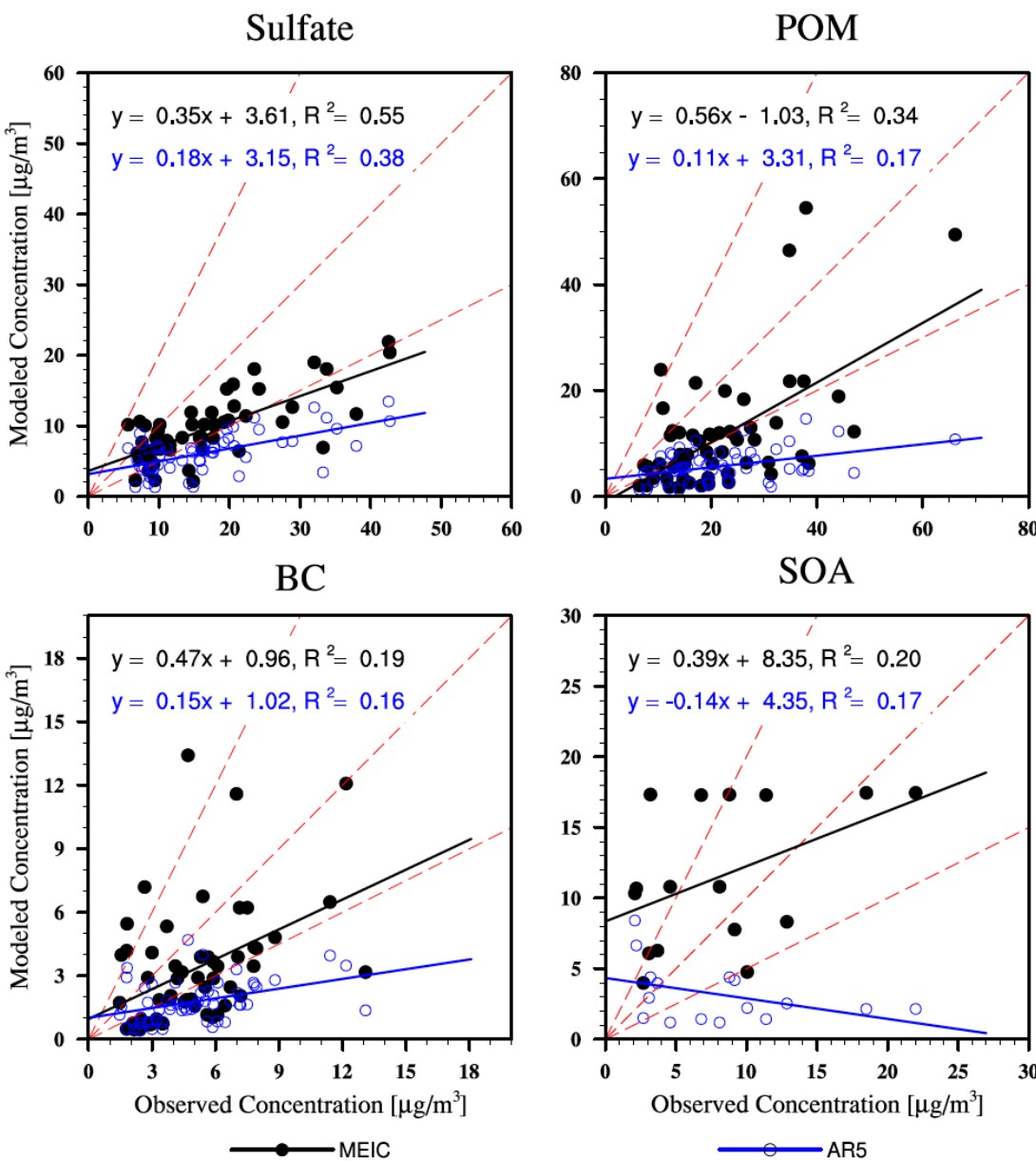

Figure 7. The monthly averaged surface concentrations of sulfate, POM, BC, and SOA using the MEIC emission and the AR5 emission compared with observations. The solid lines are linear regressions between the model results and observations. The red dashed lines represents the 1:2, 1:1, and 2:1 lines. The regression functions and coefficients of determination ($R^2$) are also shown.

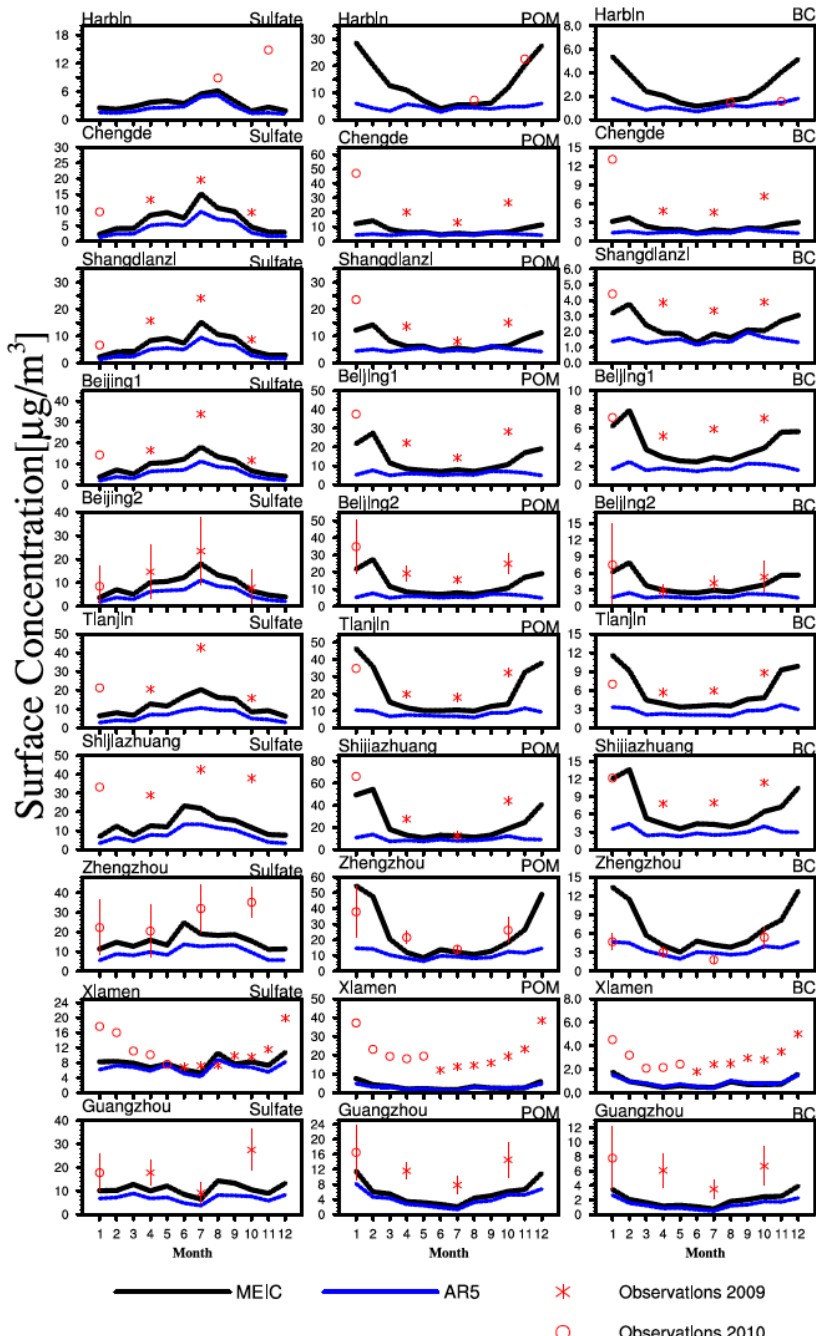

**Figure 8. The seasonal variations of monthly averaged surface concentrations of sulfate, POM, and BC modeled by CAM5-MAM3 using the MEIC (black lines) and the AR5 emissions (red lines) for year 2009 compared with the observations (asterisks for 2009 and hollow circles for 2010). Error bars stand for one standard deviation.**


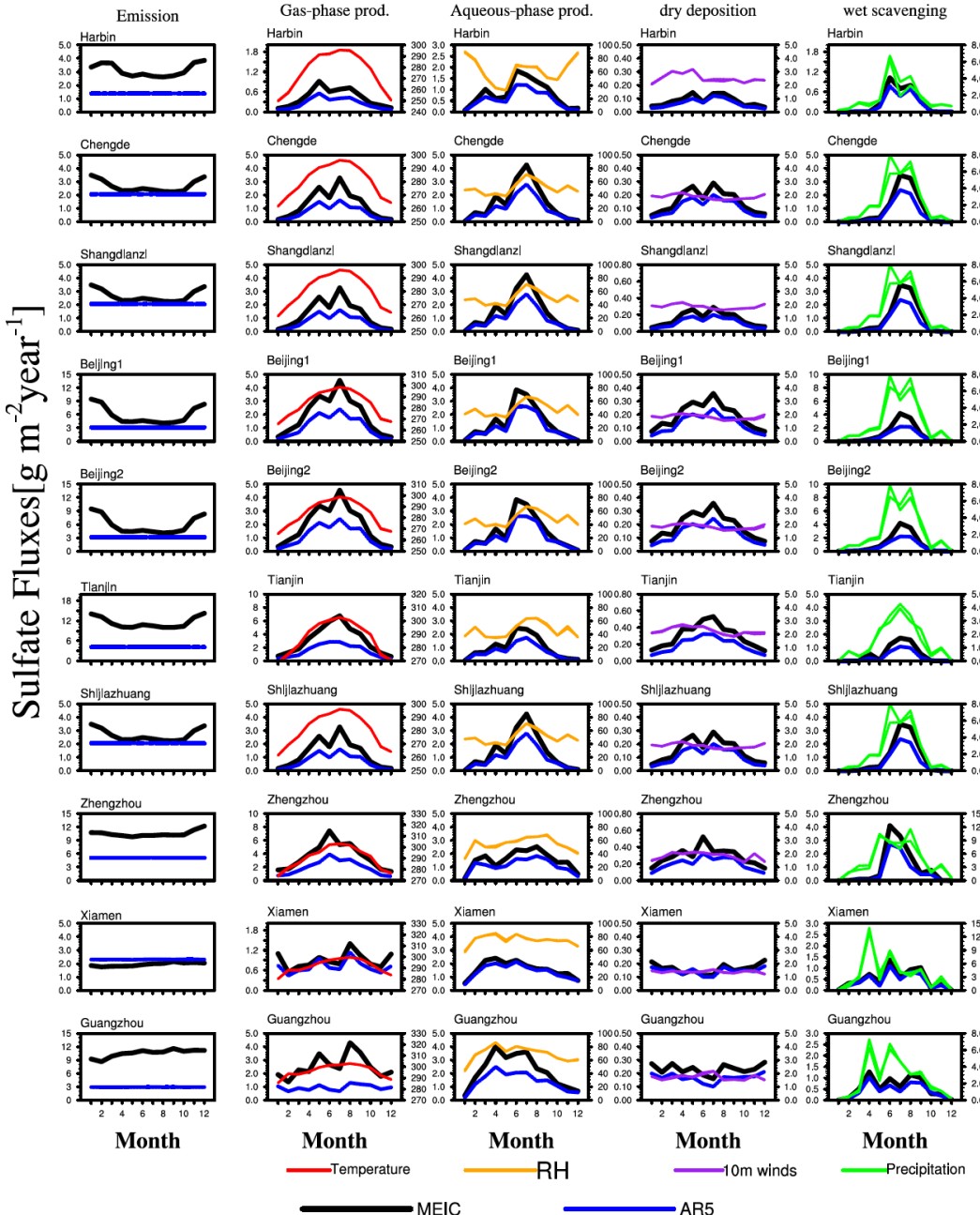

**Figure 9. From left to right columns: (1) SO₂ emission rates from the MEIC (black) and the AR5 (blue) emission inventories, model simulations for year 2009 of (2) gas-phase chemistry production rates in the simulations by MEIC and AR5 and the surface temperature (red), (3) aqueous-phase production rates and the relative humidity at surface (yellow), (4) dry deposition rates and the 10-meter wind speed (purple), and (5) wet scavenging rates and the precipitation rate (green).**


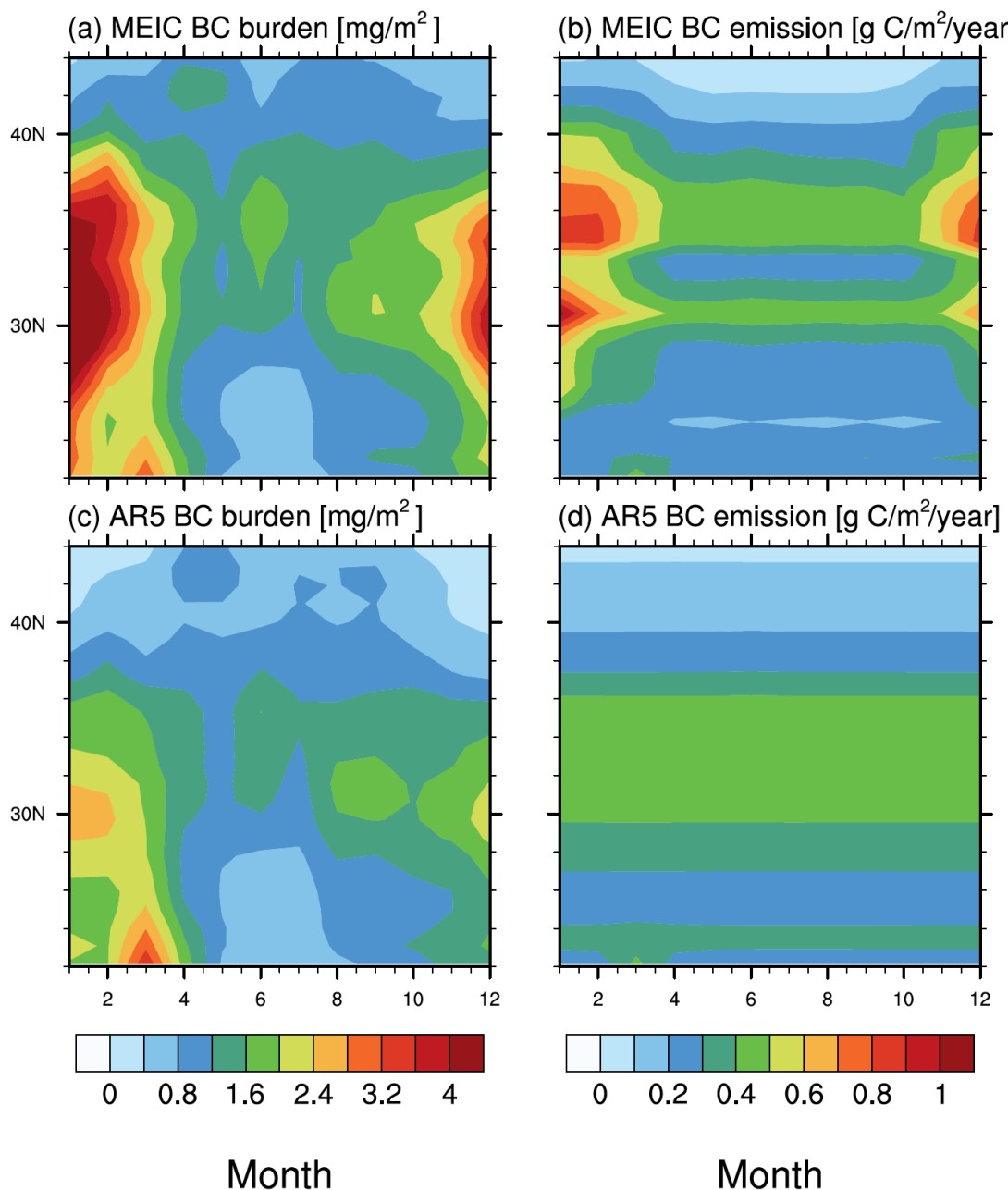

**Figure 10. The seasonal variations of longitudinal averaged (100°E-124°E) of (a) burden of BC, (b) emission rate of BC using the MEIC emission, (c) burden of BC, (d) emission rate of BC using the AR5 emission inventory over eastern China in 2009.**


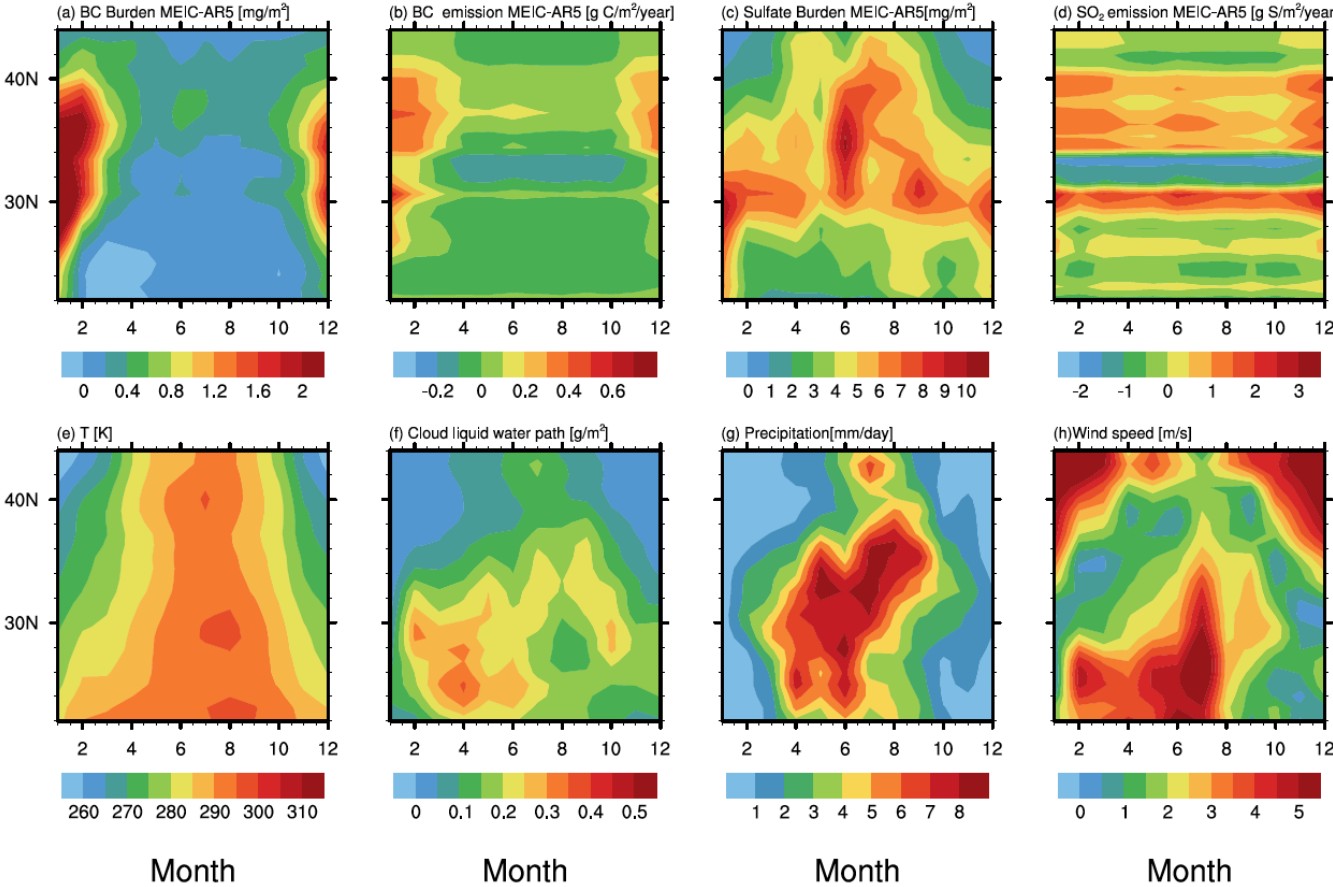

**Figure 11. The seasonal variations of longitudinal averaged ($100^o$ E-$124^o$ E) differences of (a) BC burden, (b) BC emission, (c) sulfate aerosol burden, and (d) SO$_2$ emission between the CAM5 simulations using the MEIC emission and the AR5 emission with identical meteorological variables of (e) temperature, (f) relative humidity at surface, (g) precipitation, and (h) horizontal wind speed over eastern China in 2009.**


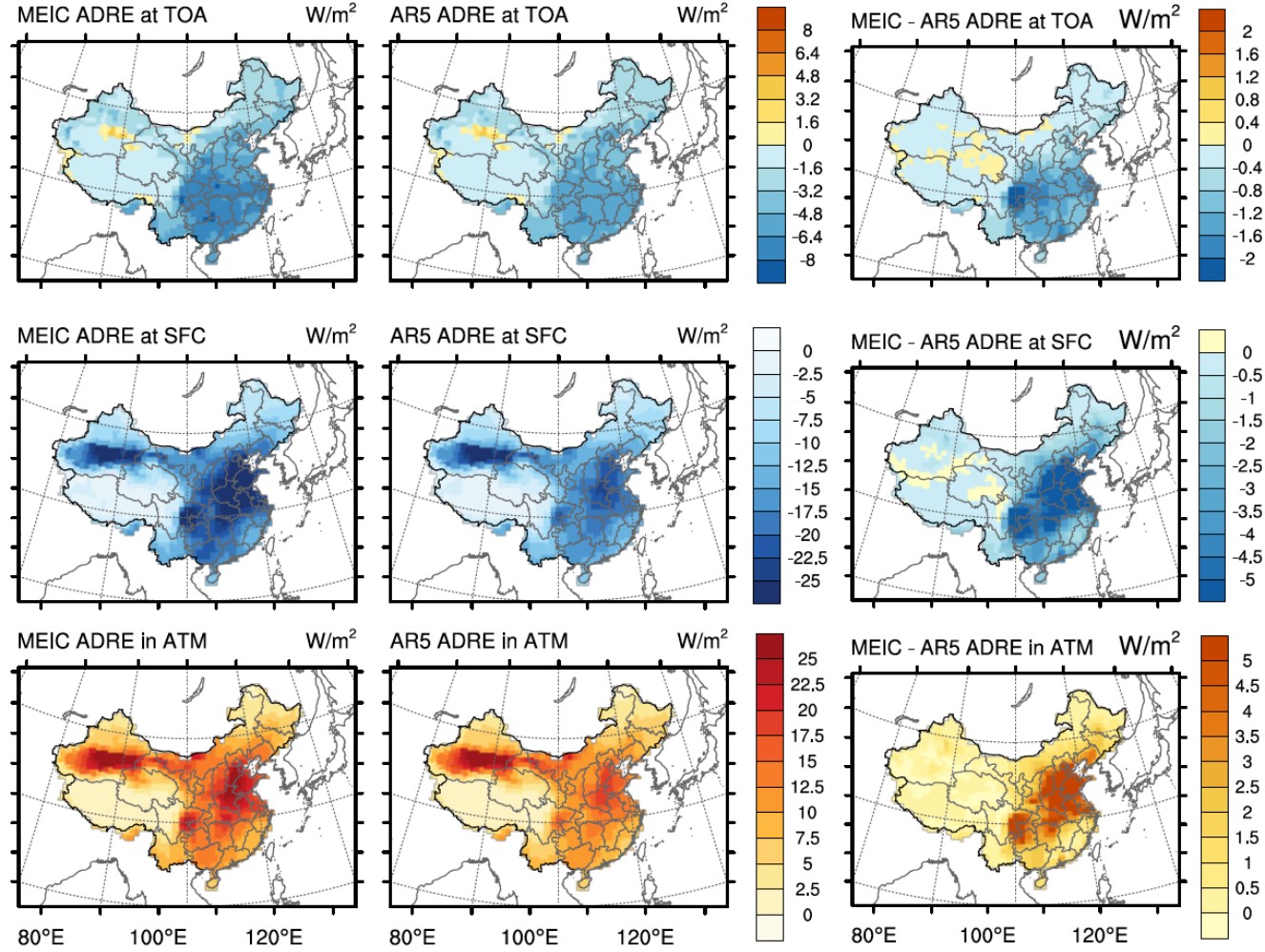

**Figure 12. Spatial distributions of the annual averaged aerosol direct radiative effects (ADREs) at TOA, surface (SFC) and in the atmosphere (ATM) using the MEIC and the AR5 emissions and their differences in year 2009.**


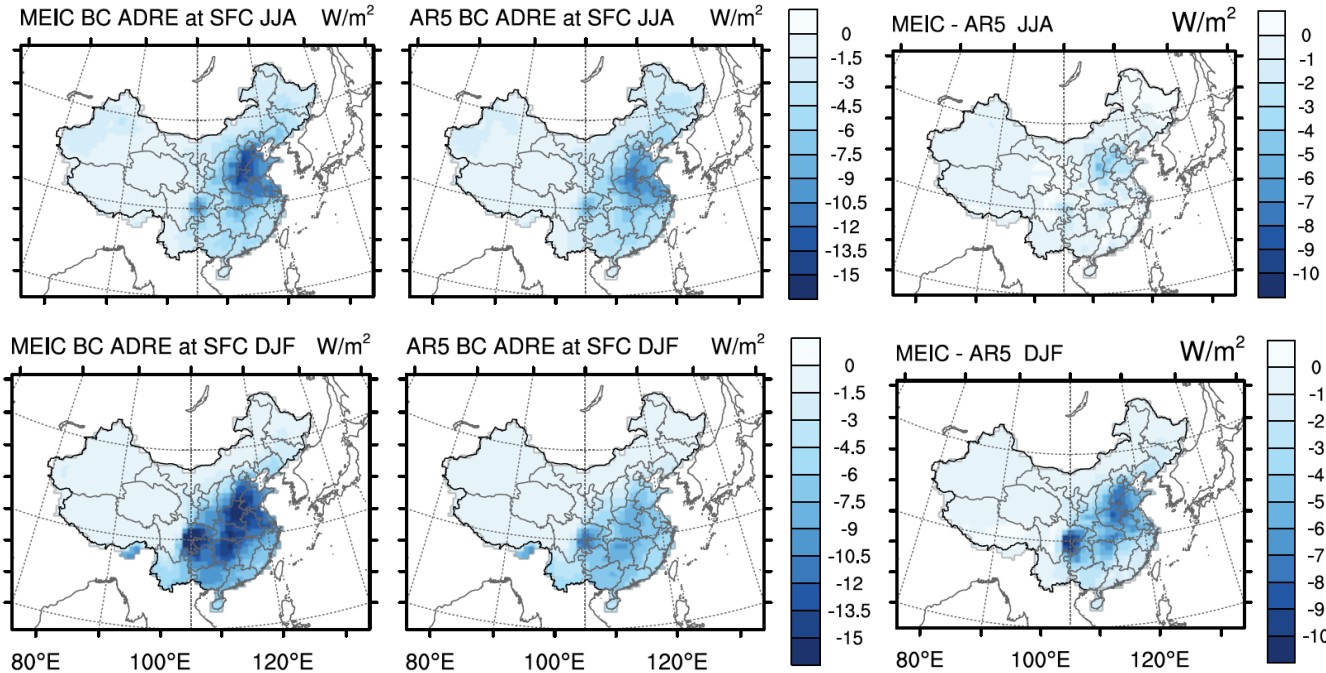

**Figure 13. Spatial distributions of ADREs of BC in summer (June, July, August) and the winter (December, January, February) at the surface (SFC) in year 2009.**


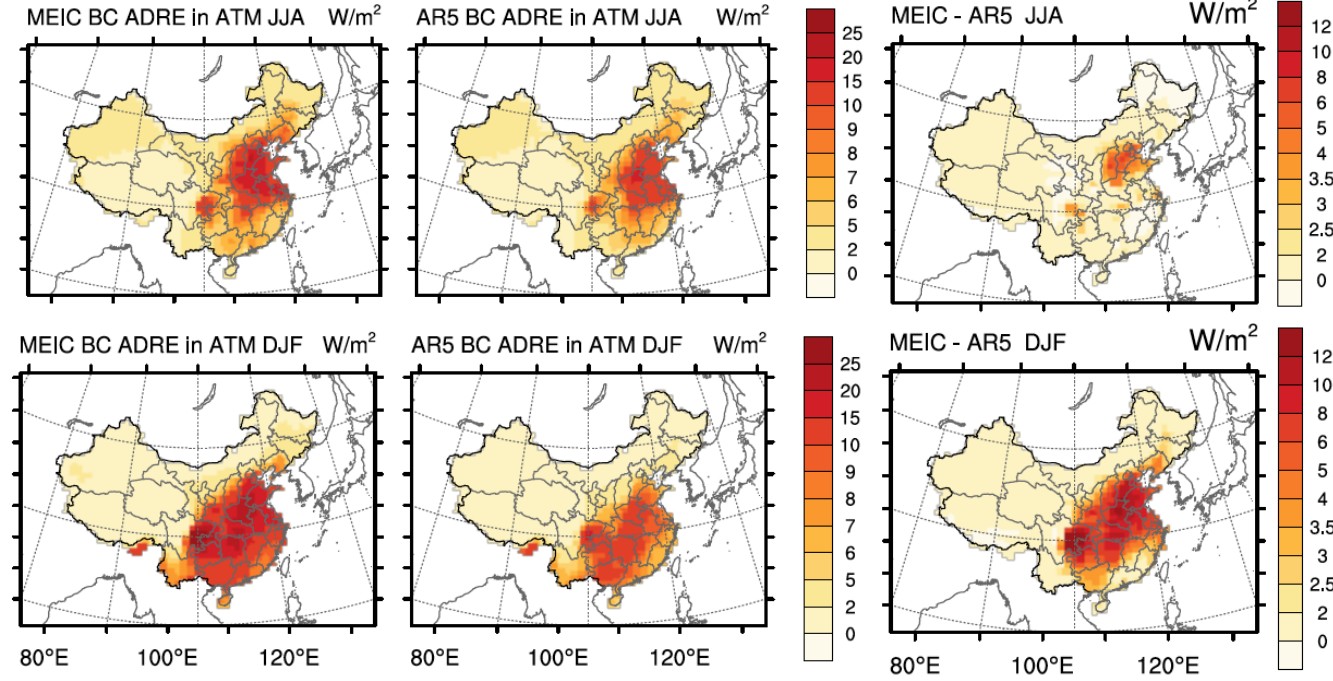


**Figure 14. Same as Figure 13 but for the ADRES of BC in the atmosphere (ATM).**

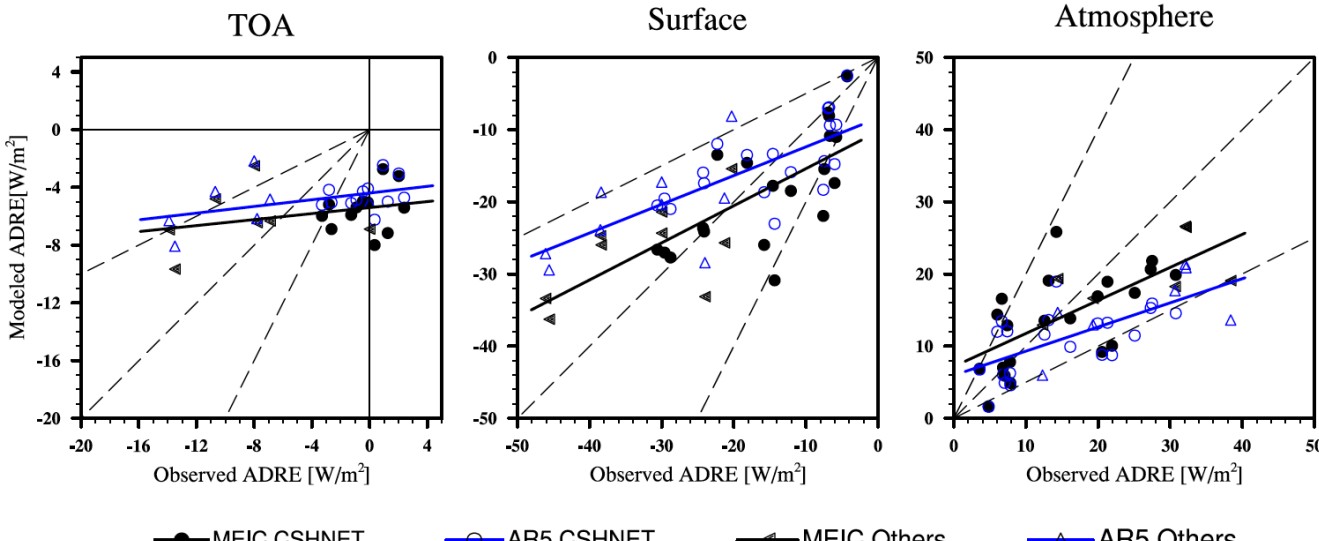

**Figure 15. ADREs at TOA, surface, and atmosphere modeled by CAM5-MAM3 using the MEIC (black dots and triangles) and the AR5 (blue dots and triangles) emissions in year 2009 compared with ADRE observations from CSHNET (dots) in year 2005 and other observations (triangles) in China for various time period ranging from 2005-2012 (Table S4) at corresponding locations. The linear regression lines between the model and the observation are also shown.**


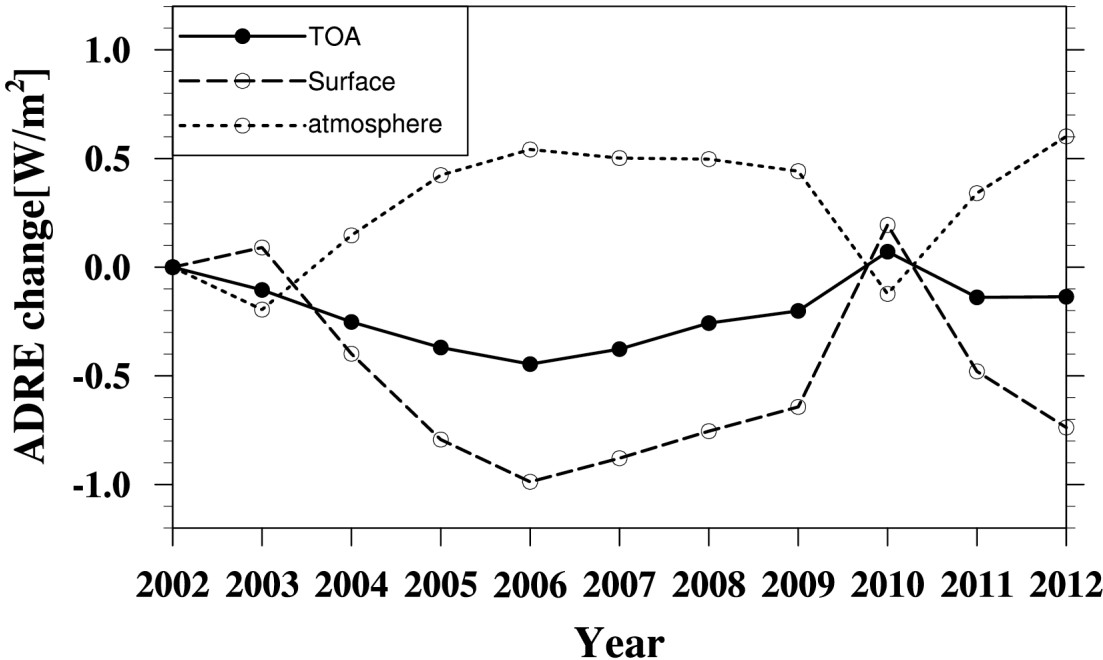

**Figure 16. The change of ADREs at TOA, surface and in the atmosphere relative to year 2002 due to the emission change from**
**2002 to 2012 in eastern China estimated by the MEIC development team.**