# Peer review of "Emission or atmospheric processes? An attempt to attribute the source of large bias of aerosols in eastern China simulated by global climate models"

_Atmospheric Chemistry and Physics, 2016_

## Referee Comment (RC1) · Anonymous Referee #1 · 20 Oct 2016

General comments

The manuscript by Fan et al. (acp-2016-802) discusses the newly emission inventories that were applied in CAM5 model. The simulated AOD, aerosol concentrations and aerosol radiative effects with the newly emissions are compared to the simulations based on IPCC AR5 emission inventories. They found the updated emissions incurred a few improvements in the simulations. The work suits the scope of the journal. The newly emission data applied in the models were appropriate and were basically clearly described. The manuscript was well written.

Unfortunately, I am having difficulty in understanding how this work in current state demonstrate evidence of novelty or advancement in the science or approach. Firstly,

the aerosol improvements due to emissions are quiet small, particularly in aerosol optical properties. This result is to some extent as expected. The author is encouraged to dig his/her own topic to extend the discussion and improve the scientific significance.

For example, China SO2 emission have been declined since 2006 while NOx and NH3 have been increasing. What's the changes in aerosol radiative forcing over China in the decade as the change in emission structure? Is it significantly different from the simulation uncertainty as using the different emissions of MEIC and IPCC AR5?

Secondly, there are large model biases in winter which cannot be explained by emissions alone and the author has no discussion. The model has a systematic bias. In fact, recently, lots of model studies claimed the importance of sulfate production through heterogeneous reactions of SO2 on deliquescence preexisting particles catalyzed by transition mental ions, which can increase PM2.5 concentrations and the mass fractions of secondary inorganic aerosols in the wintertime hazes of northern China (Wang et al. 2014; Huang et al., 2014; Zheng et al, 2015; Chen et al., 2016; Dong et al., 2016). The coexistence of NO2 and SO2 also promotes sulfate production (He et al., 2014; Wang et al., 2014). These chemical mechanisms are not appropriately considered in most models. I'm not sure that CAM5 model has well considered the sulfate chemical mechanisms in China, which is extremely likely not. The model bias should be further discussed, and the implications to the conclusion should be mentioned.

Thirdly, the observations in Figure 9 are susceptible. The observations show minimum winter sulfate in northern China cities, which is totally opposite to the general recognition that aerosols are higher in winter than in summer. An explanation is required.

Specific comments

Line 113-115, What are the emission amounts in sectors of shipping, agricultural waste burning, waste treatment and natural biomass burnings? Are they high comparing to MEIC emissions?

Line 130-140, I don't fully understand this paragraph. Did the model use BVOC from MEGAN model in MOZART-4 and use anthropogenic VOC from MEIC?

Line 140-142, Did the factor of 1.4 also apply to the natural biomass burning emissions to derive SOAG?

Line 147, Is there particle number emissions in the AR5 emission inventories?

Line 170, The simulated ADRE is a "all-sky" value while the observation-deriving ADRE in Line 187 is "clear-sky". Need to state the discrepancy.

Line 211, It might be more appropriate to show concentration results (Section 3.2) before AOD results (Section 3.3).

Line 240, why not show the results by aerosol components?

Line 249, "while the observed maximum extends further north" According to Figure 5, it seems that the observed AOD maximum is in the south of the simulated maximum. Please rewrite this sentence. Line 250-251 have the same problem.

Line 249, "This model maximum is mostly due to dust aerosol . . ." It seems that the maximum AOD in the earlier summer is about half less than the observation, and the maximum AOD is from dust. Thus, the anthropogenic AOD is quiet low comparing with MODIS. That is, CAM5 model heavily underestimates AOD in China and this cannot be explained by emissions alone. Besides to the emissions, the author need to mention other causes (e.g. missing nitrate, particle size distribution, aerosol hygroscopic growth, etc.) that account for the large AOD bias.

Line 329-330, "The sulfate concentrations in northern China . . . are characterized by the summer maximums. . ." In Figure 9, at northern cities, the minimum wintertime sulfate in observations are susceptible. As the observations were collected from literature measurements that were carried out in different periods, the observations in summer and winter may not be comparable. The comparison uncertainty should be admitted.

[Figure]

Line 343-349, in my opinion, gas-phase oxidation of SO2 is not the main pathway for sulfate production. Aqueous oxidation in droplet/cloud water should be more important. At 35°-40°N, the maximum sulfate difference between MEIC and AR5 in summer is also due to the high ambient humidity. Besides, if CAM5 can capture the wintertime high concentrations, the largest sulfate burden difference could be in winter than in summer.

Technical notes

If possible, mark the data range (i.e. Min, Max) in Figure 7 and 9.

References

Chen, D., Z. Liu, J. Fast, and J. Ban, 2016: Simulations of sulfate-nitrate-ammonium (SNA) aerosols during the extreme haze events over northern China in October 2014. Atmos. Chem. Phys., 16, 10707-10724, doi:10.5194/acp-16-10707-2016.

Dong, X., J. S. Fu, K. Huang, D. Tong, and G. Zhuang, 2016: Model development of dust emission and heterogeneous chemistry within the Community Multiscale Air Quality modeling system and its application over East Asia. Atmos. Chem. Phys., 16, 8157-8180, doi:10.5194/acp-16-8157-2016.

He, H., and Coauthors, 2014: Mineral dust and NOx promote the conversion of SO2 to sulfate in heavy pollution days. Sci. Rep., 4, 4172, doi:10.1038/srep04172.

Huang, X., Y. Song, C. Zhao, M. Li, T. Zhu, Q. Zhang, and X. Zhang, 2014: Pathways of sulfate enhancement by natural and anthropogenic mineral aerosols in China. J. Geophys. Res., 119, 14165-14179, doi:10.1002/2014JD022301.

Wang, Y., and Coauthors, 2014: Enhanced sulfate formation during China's severe winter haze episode in January 2013 missing from current models. J. Geophys. Res., 119, 10,425-10,440, doi:10.1002/2013JD021426.

Wang, Y. S., et al. (2014). "Mechanism for the formation of the January 2013 heavy

haze pollution episode over central and eastern China." Science China-Earth Sciences 57(1): 14-25.

Zheng, B., and Coauthors, 2015: Heterogeneous chemistry: a mechanism missing in current models to explain secondary inorganic aerosol formation during the January 2013 haze episode in North China. Atmos. Chem. Phys., 15, 2031-2049, doi:10.5194/acp-15-2031-2015.

---

## Referee Comment (RC2) · Anonymous Referee #2 · 13 Dec 2016

"Impact of a new emission inventory on CAM5 simulations of aerosols and aerosol radiative effects in eastern China" by Fan et al., compared the CAM5 model outputs with satellite and ground aerosol observations. The model outputs based on a new technology-based emission inventory MEIC are compared with those using IPCC AR5 emission inventory. The paper is well organized and the explanation of the experiments are clear. It is always good to have new data tested and explored to show their benefits. To be able to use new data to solve a science question is even better. This paper approves that using newly developed emission inventory, the annual mean AOD provided by CAM5 model is closer to satellite observation when compared with using old emission inventory. However, there are still scientific questions to be answered that

could make this paper more meaningful to the community.

For example, Wang et al., (2016) described a mechanism of severe haze formation in two megacities in China, which may explain part of the large differences between the CAM5 model and ground observations over Beijing and Xianghe. Also, the large differences between two inventories are within cities. The significant of this new inventory, instead of improving the annual mean AOD or altering the aerosol forcing, could be improving the regional air quality forecast.

Overall, I agree with Referee #1 that this paper needs more analyses to better understanding the model results.

References:

Wang, G., Zhang, R., Gomez, M.E., Yang, L., Zamora, M.L., Hu, M., Lin, Y., Peng, J., Guo, S., Meng, J. and Li, J., 2016. Persistent sulfate formation from London Fog to Chinese haze. ÂǎProceedings of the National Academy of Sciences, Âǎ113(48), pp.13630-13635.

---

## Author Comment (AC1) · 26 Feb 2017

We thank the referees' valuable comments and suggestions on the improving the scientific significance of this work. Here, we attempt to further explore this research topic by discussing the impact of emission uncertainty on aerosol and aerosol radiative effect (ADRE) over eastern China in the last decade, missing sources of aerosol mass in the winter, and the observed seasonality of sulfate aerosol concentration. Here are some discussions: Responds to Referee #1's comments: Referee: "China SO2 emission has been declined since 2006 while NOx and NH3 have been increasing. What's the change in aerosol radiative forcing over China in the decade as the change in emission structure? Is it significantly different from the simulation uncertainty as using the

different emissions of MEIC and IPCC AR5?" The referee addressed a very important question. We estimate the historical and future change of ADRE based on climate models. However, the uncertainty of aerosol emissions used in the climate models could add another dimension of uncertainty in simulating the change of aerosol concentration and the radiative effect. As mentioned by the referee, the emission structure changed in China during the last decade (2006-2015). The emission of SO2 decreased by 9.2% from 34.0 Tg in 2006 to 30.8 Tg in 2010 (-2.4% annual growth rate) (Lu et al., 2011). The NH3 emission decreased from 10.5 Tg in 2006 to 9.7 Tg in 2012 with an annual rate of 1.4% (Kang et al., 2016). Meanwhile, some researches show that the emissions of BC, OC, and NOx have been increasing since 2006. The BC/OC emissions increase from 1.6 Tg/3.6Tg in 2004 to 1.9 Tg/4.0 Tg in 2010, with an annual growth rate of 2.8%/1.7%. The MEIC developer team estimates that the emission of BC/OC decreased from 2006 to 2010 due to reduced emissions in the domestic and transportation sectors. The NOx emission grew by 113.9% from 12.2 Tg in 2000 to 26.1 Tg in 2010, with an annual growth rate of 7.9% from 2000-2010 (Zhao et al., 2010). To examine the change in the ADRE as the change in the emission structure, we carry out simulation using MEIC emission from 2002 to 2012. We choose these 11 years because China's economy recovered from a depression in 2002, and since then the SO2 emission start to grow dramatically and decrease after 2006 due to the application of desulfurization equipment. After 2012 the annual emission rates do not change as dramatically as the previous years. To avoid the impact of decadal trend of meteorological variations, we use the reanalysis wind in 2009 to nudge the model winds towards the "constrained meteorology" for all the years from 2002 to 2012. In this way, the change of aerosol concentrations are mostly controlled by the change of emission alone since the meteorological factors (wind speed, temperature, and etc.) are identical among the years.

Figure 1C shows the MEIC aerosol and gas precursor emission trend from 2002 to 2012 in eastern China. Since gridded MEIC emission data are only available for 2008, 2010, and 2012, we scale the spatial distribution and seasonal variation of other years

during the period to the 2008 gridded emission with the annual emissions. The annual emission rates of each species (SO2, BC, OC) are estimated by the MEIC developer team. The annual trends are consistent with other researches (Lu et al., 2011; Lei et al., 2009) although the absolute values are different. We keep using the MEIC estimations because their fuel usage is based on China Energy Statistical Yearbook (CESY) as the dataset based on the same algorithm as the gridded MEIC emission data that we used for 2009. Each species in each sector (power, energy, residential, and transportation) has a different scaling factor. The AR5 emissions from 2002 to 2009 are linearly interpolated from the only two estimates provided in the decade, that is, year 2000 and 2010. Emissions in 2011 and 2012 are set to be equal to year 2010. The uncertainty range of the MEIC SO2 emission is also shown in Figure 1C. The overall uncertainties of MEIC emissions for China in 2006 are estimated to be $\pm12\%$ for SO2, $\pm31\%$ for NOx, $\pm68\%$ for NMVOC, $\pm208\%$ for BC, and $\pm258\%$ for OC (Zhang et al., 2009). The uncertainty range is a measure of how confidence we are about the estimated emission inventory. The uncertainties for each species are calculated by the combined uncertainties of the emission measurement and the activity levels. It is measured as 95% of confidence intervals (calculated as 1.96 times coefficient of variation, which is the standard deviation divided by the mean)(Streets et al., 2003), meaning that in 95% of circumstances the SO2 emission, for example, is within $\pm12\%$ of the estimated value. The AR5 emissions of BC, OC, and SOAG are within the large uncertainty range of MEIC. However, the AR5 emission of SO2 is outside the uncertainty range of MEIC except for year 2010 to 2012. We do not scale the SOAG species since the anthropogenic VOC species have large uncertainty and are not the dominant contributor as the natural biogenic VOCs are. Unfortunately, the chemistry and aerosol microphysical processes of nitrate aerosol are not incorporated in this research. According to our latest estimation the nitrate aerosol using MEIC emission inventory improves over 60% of the underestimated AOD compared with MODIS AOD retrieval in eastern China. Figure 2C shows that the change of the ADREs from 2002 to 2012 relative to those in 2002 are 0.02~0.24 W/m2, -0.38~0.1 W/m2, and -0.07~0.57 W/m2 at TOA, surface

and in the atmosphere, respectively. The decadal trend of ADRE agrees with the trend of emissions as shown in Figure 1C. Compare with the difference between the ADREs simulated by MEIC and AR5 emission inventories in 2009, which is -0.19 W/m2, -2.42 W/m2, 2.23 W/m2 at TOA, surface and in the atmosphere, respectively, the decadal change is comparable to the uncertainty range introduced by the emission inventories. So as to answer the question by the referee, the change of ADRE in eastern China due to the change of emission structure is not significantly different than the difference estimation of the emission. It highlights the uncertainty of the emission inventories and the need of constraining the emission inventories of aerosol and precursors by in-situ and satellite observations. Our simulated decadal trend of ADRE agrees with previous findings. While decadal long observations of aerosol radiative trend over China are hard to obtain, decadal trend is often studied based on model studies. Li et al. (2014) shows that the clear-sky anthropogenic ADRE at surface increase since 2000 and peaks in the late 2000s for about -6 ∼ -7 W m-2. Westervelt et al., (2015) shows that the aerosol radiative forcing decreases by about -0.8 W m-2 in east Asia during 2005-2015 based on GFDL-CM3 model using RCP2.6 emission scenario, which shows decreasing trend of SO2 emission after 2005. Bhawar et al., (2016) shows that over East Asia smoke region the linear trend of ADRE is -0.43 W/m2/year during 2001-2012 based on CSIRO MK-3.6.0 model results.

Referee: "Secondly, there are large model biases in winter which cannot be explained by emissions alone and the author has no discussion. . . . The model bias should be further discussed, and the implications to the conclusion should be mentioned. Thirdly, the observations in Figure 9 are susceptible. The observations show minimum winter sulfate in northern China cities, which is totally opposite to the general recognition that aerosols are higher in winter than in summer. An explanation is required." The second and the third comments are about the seasonal variation of simulated aerosol concentrations, so we respond to the two commons together. As shown in Figure 3C, the emission of SO2 peaks in the winter in northern China (Harbin, Chengde, Shangdianzi, Beijing, Tianjin, Shijiazhuang, and Zhengzhou) due to heating in the domestic section.

However, the model simulates the sulfate aerosol concentrations having their minimum in the winter (Figure 9), which is commonly seen for many climate models (personal communication with Prof. Liao Hong from Nanjing University of Information Science & Technology). Obviously, as mentioned by the referee, the surface concentration cannot be explained by emissions alone. We examine the processes that determine the sulfate aerosol concentration in the model, including gas-phase and aqueous-phase production rates, the dry and wet removal rate, and the controlling meteorological variables (Figure 3C). Figure 3C shows that the simulated seasonal variation of surface concentrations of sulfate aerosol are controlled by the gas-phase and aqueous-phase production processes, as oppose to the emission rate of SO2. CAM5-MAM3 included simple gas-phase photochemical oxidation by OH and aqueous-phase oxidation in cloud water by H2O2 and O3 (Lamarque et al., 2010; Tie et al., 2001). The aerosols formed in cloud have a chance to be "resuspended" to add interstitial aerosols. Gas-phase chemistry is most active in the summer due to the temperature-dependence of the reaction rate in the photochemical oxidation of SO2 by OH. The aqueous-phase formation of sulfate aerosol also peaks in the summer due to high relative humidity. Both of gas-phase and aqueous-phase oxidations are inefficient in the winter although MEIC emission is much higher in the winter. The seasonal variation of modeled sulfate aerosol concentration is verified by some observations (Zhao et al., 2013; Zhang et al., 2013; Geng et al., 2013). However, observations from CAWNET (Zhang et al., 2012) show that sulfate aerosol in northern Chinese cities peaks in the winter as opposed to summer in spite of a minor maximum in the summer, which is consistent with our general recognition (Figure 4C). The contrasting result in observations could be due to different location and time. The difference between the simulated and the observed seasonal variation by the CAWNET may reveal that some mechanisms of sulfate aerosol formation in the winter over China are missing in the model. As indicated by the referee, "the importance of sulfate production through heterogeneous reactions of SO2 on deliquescence preexisting particles catalyzed by transition mental ions, which can increase PM2.5 concentrations and the mass fractions of secondary inorganic aerosols in the wintertime hazes of northern China (Wang et al. 2014; Huang et al., 2014; Zheng et al, 2015; Chen et al., 2016; Dong et al., 2016). The coexistence of NO2 and SO2 also promotes sulfate production (He et al., 2014; Wang et al., 2014)". The aqueous-phase oxidation of SO2 by NO2 (Wang et al., 2016) or O3 (Palout et al., 2016) is efficient to form sulfate aerosol under high relative humidity and NH3 neutralization conditions. It is also possible that the wet scavenging is underestimated in the summer so that the ADRE maximum in the summer is overestimated.

Specific comments Line 113-115, What are the emission amounts in sectors of shipping, agricultural waste burning, waste treatment and natural biomass burnings? Are they high comparing to MEIC emissions? The emission amounts in sectors of shipping, agricultural waste burning, waste treatment and natural biomass burning are not included in MEIC emission. Therefore we keep them the same as the AR5 emission.

Line 130-140, I don't fully understand this paragraph. Did the model use BVOC from MEGAN model in MOZART-4 and use anthropogenic VOC from MEIC? Yes, the referee's understanding is basically correct. We rewrite the description in line as follows: "Since the IPCC AR5 dataset does not provide emissions of biogenic VOCs, the SOAG emission is derived from the emission fluxes of five primary VOC categories (isoprene, monoterpenes, big alkanes, big alkenes, toluene) that are prescribed from the Model for OZone And Related chemical Tracers version 4 (MOZART-4) dataset (Emmons et al., 2010), in which the biogenic emissions of isoprene and monoterpenes are based on the Model for Emissions of Gases and Aerosols Emissions from Nature (MEGAN) (Guenther et al., 2006). The MEIC emission provides anthropogenic sources of the five VOC categories and the mapping table for lumping the MEIC VOC species to MOZART refers to Li et al. (2014). Since the MEIC does not provide the biogenic sources of isoprene and monoterpenes, which are much larger than their natural counterparts, we make their total emissions from anthropogenic and natural sources the same as those in the AR5 emission.

Line 140-142, Did the factor of 1.4 also apply to the natural biomass burning emissions

to derive SOAG? Yes, it does. When deriving SOAG emission, all anthropogenic and biomass burning sectors are added up and multiplied by 1.4 to convert OC (carbon mass) to POM (organic mass).

Line 147, Is there particle number emissions in the AR5 emission inventories? No, the particle number emission in the AR5 emission inventory is not provided. The number concentration in both AR5 and MEIC emissions are calculated from the mass concentration as described in the text. To be more clear, the paragraph is rewritten as follows: "The number emission fluxes in both AR5 and MEIC are calculated from the mass fluxes in a consistent way. The mass to number conversion is based on E_number=E_mass/(($\pi$/6 ãĂŰD_vãĂŮ^3 ) ), where Dv is the volume-mean emitted diameter and  is the aerosol particle density (Liu et al., 2012). Since the MEIC emission does not provide mass emissions from agricultural waste burning, waste treatment, forest fire, grass fire and continuous volcanoes, we use the number fluxes from the AR5 emission for these sectors."

Line 170, The simulated ADRE is a "all-sky" value while the observation-deriving ADRE in Line 187 is "clear-sky". Need to state the discrepancy. The cloud-screening method (clear-sky) in observations neglects the effect of aerosol above and below clouds. Absorption of reflected solar radiation by absorbing aerosol above clouds should result in larger radiative warming effect. The radiative cooling by scattering aerosol below clouds is not as strong as in clear sky due to less sunlight penetrating the clouds. Both of the two factors will results in lower (more negative at TOA) radiative effect in clear-sky than that in all-sky. Therefore, the observation-derived ADRE in clear-sky provides an upper-limit of the model- estimated all-sky ADRE at TOA.

Line 211, It might be more appropriate to show concentration results (Section 3.2) before AOD results (Section 3.3). Thank you for the suggestion. It is more conventional to show surface concentration before AOD. Nevertheless, we show AOD first since we would like to address the question of whether the underestimated AOD by models is improved by using MEIC emission. Also, since satellite retrieved AOD provides the

spatial distribution to be compared with the model results, we think it might be acceptable to show AOD before concentration results.

Line 240, why not show the results by aerosol components? The AOD by aerosol components has been added in the appendix.

Line 249, "while the observed maximum extends further north" According to Figure 5, it seems that the observed AOD maximum is in the south of the simulated maximum. Please rewrite this sentence. Line 250-251 have the same problem. We delete rewrite the sentences as "The model simulates a maximum between 35oN and 40oN in early summer (from May to July) which is to the north of the observed AOD maximum.".

Line 249, "This model maximum is mostly due to dust aerosol . . ." It seems that the maximum AOD in the earlier summer is about half less than the observation, and the maximum AOD is from dust. Thus, the anthropogenic AOD is quiet low comparing with MODIS. That is, CAM5 model heavily underestimates AOD in China and this cannot be explained by emissions alone. Besides to the emissions, the author need to mention other causes (e.g. missing nitrate, particle size distribution, aerosol hygroscopic growth, etc.) that account for the large AOD bias. We agree with the referee that other causes beside emission also account for the large AOD bias. We mentioned the plausible causes and future effort to be made in Line 258-262 and the summary section (Line 443-449). We rewrite the sentences as "This model maximum is mostly due to dust aerosol transported from the west, while the satellite retrievals do not show such strong dust emission and transport. The timing and location of dust aerosol in the model is biased. Since the dust emissions are the same in the simulations using the MEIC and AR5 emissions, the difference of the modeled AOD maximums between 35oN and 40oN is mainly due to sulfate condensed on dust aerosol. We notice that the maximum AOD in the satellite retrieval occurs around 30oN in early summer (May to July), as oppose to 35oN to 40oN as simulated by the model. The observed AOD maximum complies with the SO2 emission maximum in early summer around 30oN. Therefore, it is likely that this AOD maximum is due to sulfate aerosol formation and

it is heavily underestimated by the model. Since the uncertainty of SO2 emission is relative low (±12%), this underestimation cannot be explained by emission alone. "

Line 329-330, "The sulfate concentrations in northern China are characterized by the summer maximums" In Figure 9, at northern cities, the minimum wintertime sulfate in observations are susceptible. As the observations were collected from literature measurements that were carried out in different periods, the observations in summer and winter may not be comparable. The comparison uncertainty should be admitted. The summer maximums of the surface concentration of sulfate at northern cities in 2009 and 2010 are observed by some independent researches (Zhao et al., 2013; Zhang et al., 2013; Geng et al., 2015). It is possible that SO2 to sulfate conversion is efficient in the summer due to high temperature and relative humidity. There are also observations showing opposite seasonal variations with sulfate concentration peaking in the winter during other periods (Zhang et al., 2012). The contrasting result in the observations could be due to different location and time.

Line 343-349, in my opinion, gas-phase oxidation of SO2 is not the main pathway for sulfate production. Aqueous oxidation in droplet/cloud water should be more important. At 35o-40oN, the maximum sulfate difference between MEIC and AR5 in summer is also due to the high ambient humidity. Besides, if CAM5 can capture the wintertime high concentrations, the largest sulfate burden difference could be in winter than in summer. We totally agree with the referee. We add the seasonal variation of relative humidity in Figure 11. One thing to notice is that aqueous-phase oxidation occurs only in cloud droplets in the model. The aqueous-phase oxidation of SO2 by NO2 on pre-existing aerosols is not modeled by CAM5-MAM3, which is important in China (Wang et al., 2016). The model results show that the aqueous-phase oxidation is as important as the gas-phase chemistry. If we consider the oxidation by NO2 on pre-existing aerosols, the contribution of aqueous-phase chemistry could be more important. Technical notes If possible, mark the data range (i.e. Min, Max) in Figure 7 and 9. Data range (one standard deviation) in Figure 7 and Figure 9 are marked. The standard deviations of

the surface concentrations are provided in only three locations (Beijing, Zhengzhou, and Guangzhou) by the literatures. We add them to Figure 9. Figure 9. Error bars stand for one standard deviations of the observations. Figure captions:

Figure 1C. The change of emission rates of SO2, BC, OC, and SOAGs from year 2002 to 2012 in eastern China. The shaded areas are the uncertainty ranges of MEIC SO2 emission. The uncertainty ranges of POM, BC, and SOAG ($\pm256\%$, $\pm208\%$, and $\pm68\%$, respectively) fulfill the plot range.

Figure 2C. The change of ADREs at TOA, surface and in the atmosphere relative to year 2002 with MEIC emission in eastern China.

Figure 3C (From left to right columns): (1) SO2 emission rates from MEIC (black) and AR5 (blue) inventories, (2) gas-phase chemistry production in the simulations by MEIC (black) and AR5 (blue) and the surface temperature (red), (3) aqueous-phase production rates (black and blue) and the relative humidity at surface (red), (4) dry deposition rate (black and blue) and the 10-meter wind speed (red), and (5) wet scavenging rate (black and blue) and the precipitation rate (red).

Figure 4C. Seasonal variations of surface sulfate concentration at three locations (Gucheng, Panyu, and Zhengzhou) from CAWNET from 2006 to 2007. For comparison, the observations near the three locations used in our study (Beijing, Guangzhou, Zhengzhou) from 2009 to 2010 are also shown in dots.

Figure S1. The seasonal variation of longitudinal averaged (100oE-124oE) AOD at 550 nm by aerosol components (dust, sulphate, BC, POM, and SOA from top to bottom) simulated by CAM5-MAM3 using the MEIC emission (left column) and the AR5 emission (the right column).

Figure 6C. The seasonal variation of longitudinal averaged SO2 emission from MEIC [g S/m2/year].

Revised Figure 11.

Revised Figure 7. Error bars stand for one standard deviations of the observations. Revised Figure 9. Error bars stand for one standard deviations of the observations.

References: Bhawar, R. L., Lee, W.-S., and Pahu,l P.R.C.: Aerosol types and radiative forcing estimates over East Asia, Atmos. Environ., 141, 532-541, doi: 10.1016/j.atmosenv.2016.07.028, 2016. Chen, D., Liu, Z., Fast, J., and Ban, J.: Simulations of sulfate-nitrate-ammonium (SNA) aerosols during the extreme haze events over northern China in October 2014. Atmos. Chem. Phys., 16, 10707-10724, doi:10.5194/acp-16-10707-2016, 2016. Dong, X., Fu, J. S., Huang, K., Tong, D., and Zhuang, G.: Model development of dust emission and heterogeneous chemistry within the Community Multiscale Air Quality modeling system and its application over East Asia. Atmos. Chem. Phys., 16, 8157-8180, doi:10.5194/acp-16-8157-2016, 2016. Geng, N., Wang, J., Xu, Y., Zhang, W., Chen, C., and Zhang, R.: PM2.5 in an industrial district of Zhengzhou, China: chemical composition and source apportionment, Particuology, 11(1), 99-109, 2013. He, H., Wang, Y., Ma, Q., Ma, J., Chu, B., Ji, D., Tang, G., Liu, C. Zhang, H., and Hao, J.: Mineral dust and NOx promote the conversion of SO2 to sulfate in heavy pollution days. Sci. Rep., 4, 4172, doi:10.1038/srep04172, 2014. Huang, X., Song, Y., Zhao, C., Li, M., Zhu, T., Zhang, Q., and Zhang, X.: Pathways of sulfate enhancement by natural and anthropogenic mineral aerosols in China. J. Geophys. Res., 119, 14165-14179, doi:10.1002/2014JD022301, 2014. Lu, Z., Zhang Q., and Streets D. G.: Sulfur dioxide and primary carbonaceous aerosol emissions in China and India, 1996–2010 Atmos. Chem. Phys., 11, 9839–9864, doi:10.5194/acp-11-9839-2011, 2011. Li, Jiandong, Wei-Chyung Wang, Zhian Sun, Guoxiong Wu, Hong Liao, Yimin Liu, Decadal variation of East Asian radiative forcing due to anthropogenic aerosols during 1850–2100, and the role of atmospheric moisture, Clim Res, Vol. 61: 241–257, doi: 10.3354/cr01236, 2014. Paulot F, Fan S, Horowitz L W.: Contrasting seasonal responses of sulfate aerosols to declining SO2 emissions in the Eastern US: implications for the efficacy of SO2 emission controls, Geophys. Res, Lett., 2016. Streets, D. G., Bond, T. C., Carmichael, G. R., Fernandes, S. D., Fu, Q., He, D., Klimont, Z., Nelson, S. M., Tsai, N. Y., Wang, M. Q., Woo, J.-H., and

Yarber, K. F.: An inventory of gaseous and primary aerosol emissions in Asia in the year 2000, J. Geophys. Res., 108, 8809, doi:10.1029/2002JD003093, 2003. Wang, Y., Zhang, Q., Jiang, J., Zhou, W.: Enhanced sulfate formation during China's severe winter haze episode in January 2013 missing from current models. J. Geophys. Res., 119, 10,425-10,440, doi:10.1002/2013JD021426, 2014. Wang, Y. S., Yao, L., Wang, L., Liu, Z., et al.: Mechanism for the formation of the January 2013 heavy haze pollution episode over central and eastern China." Science China-Earth Sciences, 57(1): 14-25, 2014. Wang G, Zhang R, Gomez M E, et al. :Persistent sulfate formation from London Fog to Chinese haze., Proceedings of the Nat. Acad. of Sci., 201616540, 2016. Westervelt, D. M., Horowitz, L.W., Naik. V., et al: Radiative forcing and climate response to projected 21st century aerosol decreases. Atmos. Chem. and Phys., 15(22): 12681-12703, 2015. Zhang, R., Jing, J., Tao, J., Hsu, S.-C., Wang, G., Cao, J., Lee, C. S. L., Zhu, L., Chen, Z., Zhao, Y., and Shen, Z.: Chemical characterization and source apportionment of PM2.5 in Beijing: seasonal perspective, Atmos. Chem. Phys., 13(14), 7053-7074, 2013. Zhao, P. S., Dong, F., He, D., Zhao, X. J., Zhang, X. L., Zhang, W. Z., Yao, Q., and Liu, H. Y.: Characteristics of concentrations and chemical compositions for PM2.5 in the region of Beijing, Tianjin, and Hebei, China, Atmos. Chem. Phys., 13, 4631-4644, 2013. Zheng, B., Zhang, Q., Zhang, Y., He, K. and et al.: Heterogeneous chemistry: a mechanism missing in current models to explain secondary inorganic aerosol formation during the January 2013 haze episode in North China. Atmos. Chem. Phys., 15, 2031-2049, doi:10.5194/acp-15-2031-2015, 2015.

[Figure]

**Fig. 1.** Fig 1C

[Figure]

**Fig. 2.** Fig 2C

[Figure]

**Fig. 3.** Fig 3C

[Figure]

[Figure: line and scatter plot of Sulfate surface concentration (µg/m3) versus Month, with legend showing Gucheng 2006-2007, Panyu 2006-2007, Zhengzhou 2006-2007, Beijing (Zhang) 2009-2010, Guangzhou 2009-2010, Zhengzhou 2010, Beijing (Zhao) 2009-2010]

**Fig. 4.** Fig 4C

[Figure]

**Fig. 5.** Fig S1.
MEIC SO2

Fig. 6. Fig 6C

[Figure]

**Fig. 7.** Revised Fig 11

[Figure]

**Fig. 8.** Revised Fig 9

---

## Author Comment (AC2) · 26 Feb 2017

Responds to Referee #2's comments: We thank the referees' valuable comments and suggestions on the improving the scientific significance of this work.

Referee: "However, there are still scientific questions to be answered that could make this paper more meaningful to the community. For example, Wang et al., (2016) described a mechanism of severe haze formation in two megacities in China, which may explain part of the large differences between the CAM5 model and ground observations over Beijing and Xianghe." We totally agree with the referee that the difference between the CAM5 model and ground observations over Beijing and Xianghe may be explained by a missing mechanism of the aqueous-phase reaction in neutralized haze

particles, which is proved to be of critical effects on the formation of severe haze in China (Wang et al., 2016). In order to characterize the aerosol properties in China, more work needs to be done to include the aqueous-phase chemistry in the aerosol particles, as well as cloud droplets. Our improved version of the chemistry and aerosol mechanism of the CAM5 model incorporates nitrate and ammonium chemistry, which provides a basis for this further examination of the proposed missing source of aerosol in the model. Referee: "Also, the large differences between two inventories are within cities. The significant of this new inventory, instead of improving the annual mean AOD or altering the aerosol forcing, could be improving the regional air quality forecast. Overall, I agree with Referee #1 that this paper needs more analyses to better understanding the model results." We agree with the referee that the significant of the new inventory, MEIC, could improve the regional air quality forecast. Actually, MEIC has been applied to air quality models (i.e., CMAQ, Liu et al., 2010) and The advantage of MEIC emission in characterizing the seasonal variation of emission in China will show its significance in regional air quality forecast in different seasons. In addition to improve the annual mean AOD (Figure 4) and ADRE (Figure 12), the MEIC inventory has shown its advantage in modeling the seasonal variation of AOD (Figure 5) and surface concentration of primary aerosols (BC and POM, Figure 9). By introducing the nitrate and ammonium chemistry in aerosol particles, the seasonal variations of secondary aerosols (e.g., sulfate) are also expected to improve.

---

## Author Comment (AC3) · 9 Mar 2017

We thank the referees' valuable comments and suggestions on the improving the scientific significance of this work. Here, we attempt to further explore this research topic by discussing the impact of emission uncertainty on aerosol and aerosol radiative effect (ADRE) over eastern China in the last decade, missing sources of aerosol mass in the winter, and the observed seasonality of sulfate aerosol concentration. Here are some discussions:

**Responds to Referee #1's comments:**

**Referee: "China SO$_2$ emission has been declined since 2006 while NO$_x$ and NH$_3$ have been increasing. What's the change in aerosol radiative forcing over China in the decade as the change in emission structure? Is it significantly different from the simulation uncertainty as using the different emissions of MEIC and IPCC AR5?"**

The referee raised an important question. We rely on climate models to estimate the historical and future changes of ADRE. However, the uncertainty of aerosol emissions used in the climate models could add another dimension of uncertainty in simulating the change of aerosol radiative effect. As mentioned by the referee, the emission structure changed in China during the last decade (2006-2015). The emission of SO$_2$ decreased by 9.2% from 34.0 Tg in 2006 to 30.8 Tg in 2010 (-2.4% annual growth rate) (Lu et al., 2011). The NH$_3$ emission decreased slightly from 10.5 Tg in 2006 to 9.7 Tg in 2012 with an annual rate of 1.4% (Kang et al., 2016). Meanwhile, some researches show that the emissions of BC, OC, and NO$_x$ have been increasing since 2006. The BC/OC emissions increase from 1.6 Tg/3.6Tg in 2004 to 1.9 Tg/4.0 Tg in 2010, with an annual growth rate of 2.8%/1.7%. The MEIC developer team estimates that the emission of BC/OC decreased from 2006 to 2010 due to reduced emissions in the domestic and transportation sectors. The NO$_x$ emission grew by 113.9% from 12.2 Tg in 2000 to 26.1 Tg in 2010, with an annual growth rate of 7.9% from 2000-2010 (Zhao et al., 2010).

To examine the change in the ADRE as the change in the emission structure, we carry out a simulation using MEIC emission from 2002 to 2012. We choose these 11

years because China's economy recovered from a depression in 2002, and since then the SO₂ emission has started to grow dramatically and has been decreasing after 2006 due to the application of desulfurization equipment. After 2012 the annual emission rates do not change as dramatically as the previous years. To separate from the impact of decadal variation of meteorological variations, we use the reanalysis wind in 2009 to nudge the model meteorological fields (winds, temperature, and etc.) towards the "constrained meteorology" for this simulation. In this way, the change of aerosol concentration is controlled by the change of emission alone since the meteorological fields are identical among the years.

[Figure]

Figure S1. The change of emission rates of SO₂, BC, and OC from year 2002 to 2012 in eastern China.

Figure S1 shows the MEIC 's SO₂, BC, and OC emission trends from 2002 to 2012 in eastern China. Since the spatially-gridded MEIC emission data are only available for 2008, 2010, and 2012, we scale the spatial distribution and seasonal variation of other years during the period to the 2008 gridded emission with the annual emissions. Each species in different sectors (power, energy, residential, and transportation) has a different scaling factor. The annual emission rates are estimated by the MEIC developer team. The annual trends are consistent with other researches (Lu et al., 2011; Lei et al., 2009) although the absolute values are different. We use

the MEIC estimations because their fuel usage is based on China Energy Statistical Yearbook (CESY) as this dataset is based on the same algorithm as the gridded MEIC emission data that we used for 2009.

[Figure]

Figure S2. The change of ADREs at TOA, surface and in the atmosphere relative to year 2002 with MEIC emission in eastern China.

Figure S2 shows that the change of the ADREs from 2002 to 2012 relative to those in 2002 are 0.01~0.16 $W/m^2$, -0.44~0.39 $W/m^2$, and -0.23~0.51 $W/m^2$ at TOA, surface and in the atmosphere, respectively. The decadal trend of ADRE agrees with the trend of emissions as shown in Figure S1. The warming in the atmosphere and the cooling at the surface enhanced with the increase of emissions of $SO_2$, BC, and POM from 2003 to 2006. The ADRE at TOA increases slightly indicating more energy retains in the atmosphere-earth system. From 2006 to 2009, the changes of ADREs are not significant due to the stabilized emission of BC. Since 2010, the warming in the atmosphere and the cooling at the surface increases due to the increase emission of $SO_2$ and BC. The changes of ADREs at surface and in the atmosphere from 2002 to 2003 may reflect the complicated interactions between sulfate and BC/OC in East China, enhancing the BC/OC wet scavenging due to sulfate coating.

Compare with the difference between the ADREs simulated by MEIC and AR5

emission inventories in 2009, which is -0.19 W/m$^2$, -2.42 W/m$^2$, 2.23 W/m$^2$ at TOA, surface and in the atmosphere, respectively, the decadal change of ADRE at TOA is comparable to the uncertainty range introduced by the emission inventories. The decadal changes of ADRE at surface and in the atmosphere are smaller than the uncertainties by using the two different emissions. So as to answer the question by the referee, the change of ADRE at TOA in eastern China due to the change of emission structure is not significantly different than the difference estimation of the emission, although the magnitude of changes at surface and in the atmosphere is smaller. It highlights the uncertainty of the emission inventories and the need of constraining the emission inventories of aerosol and precursors by in-situ and satellite observations.

**Referee: "Secondly, there are large model biases in winter which cannot be explained by emissions alone and the author has no discussion. ... The model bias should be further discussed, and the implications to the conclusion should be mentioned.**

**Thirdly, the observations in Figure 9 are susceptible. The observations show minimum winter sulfate in northern China cities, which is totally opposite to the general recognition that aerosols are higher in winter than in summer. An explanation is required."**

The second and the third comments are about the seasonal variations of simulated aerosol concentrations, so we respond to the two commons together. As shown in Figure S3, the emission of SO$_2$ peaks in the winter in northern Chinese cities (Harbin, Chengde, Shangdianzi, Beijing, Tianjin, Shijiazhuang, and Zhengzhou) due to heating in the domestic sector. However, the modeled sulfate aerosol concentrations show their minimum in the winter (Figure 9), which is commonly seen for many climate models (personal communication with Prof. Liao Hong from Nanjing University of Information Science & Technology). Obviously, as mentioned by the referee, the surface concentration cannot be explained by emissions alone, and atmospheric physical and chemical processes are more responsible for the low bias. We examine the processes that determine the sulfate aerosol concentration in the model, including

gas-phase and aqueous-phase production, the dry removal and wet scavenging, as well as the controlling meteorological variables (Figure S3).

[Figure]

Figure S3. (From left to right columns): (1) SO$_2$ emission rates from MEIC (black) and AR5 (blue) inventories, (2) gas-phase chemistry production in the simulations by MEIC (black) and AR5 (blue) and the surface temperature (red), (3) aqueous-phase production rates (black and blue) and the relative humidity at surface (yellow), (4) dry

deposition rate (black and blue) and the 10-meter wind speed (purple), and (5) wet scavenging rate (black and blue) and the precipitation rate (green).

Figure S3 shows that the simulated seasonal variations of surface concentrations of sulfate aerosol are controlled by the gas-phase and aqueous-phase production processes, as oppose to the emission of $SO_2$. Gas-phase chemistry is most active in the summer due to the temperature-dependence of the reaction rate in the photochemical oxidation of $SO_2$ by OH. The aqueous-phase formation of sulfate aerosol also peaks in the summer due to high relative humidity. Although the MEIC emission of $SO_2$ peak in the winter, both gas-phase and aqueous-phase oxidations are inefficient in the winter, which results in lower concentrations of sulfate aerosol than in the summer. The seasonal variation of modeled sulfate aerosol concentration is verified by some observations (Zhao et al., 2013; Zhang et al., 2013; Geng et al., 2013). However, observations from CAWNET (Zhang et al., 2012) show that sulfate aerosol in northern Chinese cities (Gucheng and Zhengzhou in Figure S4) peaks in the winter as opposed to summer in spite of a minor maximum in the summer.

[Figure]

Figure S4. Seasonal variations of surface sulfate concentration at three locations (Gucheng, Panyu, and Zhengzhou) from CAWNET from 2006 to 2007. For

comparison, the observations near the three locations used in our study (Beijing, Guangzhou, Zhengzhou) from 2009 to 2010 are also shown in dots.

The contrasting result in observations could be due to different location and time. The difference between the simulated and the observed seasonal variations by the CAWNET may reveal that some mechanisms of sulfate aerosol formation in the winter over China are missing in the model. As indicated by the referee, "the importance of sulfate production through heterogeneous reactions of $SO_2$ on deliquescence preexisting particles catalyzed by transition mental ions, which can increase PM2.5 concentrations and the mass fractions of secondary inorganic aerosols in the wintertime hazes of northern China (Wang et al. 2014; Huang et al., 2014; Zheng et al, 2015; Chen et al., 2016; Dong et al., 2016). The coexistence of $NO_2$ and $SO_2$ also promotes sulfate production (He et al., 2014; Wang et al., 2014)". The aqueous-phase oxidation of $SO_2$ by $NO_2$ (Wang et al., 2016) or $O_3$ (Palout et al., 2016) is efficient to form sulfate aerosol under high relative humidity and $NH_3$ neutralization conditions.

Following the referee's comments, we have added the above discussion in the revised manuscript, including the missing of sulfate production mechanisms in CAM5 model.

**Specific comments**

**Line 113-115, What are the emission amounts in sectors of shipping, agricultural waste burning, waste treatment and natural biomass burnings? Are they high comparing to MEIC emissions?**

The emission amounts in sectors of shipping, agricultural waste burning, waste treatment and natural biomass burning are not included in the MEIC emission. Therefore we keep them the same as the AR5 emission.

**Line 130-140, I don't fully understand this paragraph. Did the model use BVOC from MEGAN model in MOZART-4 and use anthropogenic VOC from MEIC?**

Yes, the referee's understanding is correct. We rewrite the description in line as follows:

"Since the IPCC AR5 dataset does not provide emissions of biogenic VOCs, the SOAG emission is derived from the emission fluxes of five primary VOC categories (isoprene, monoterpenes, big alkanes, big alkenes, toluene) that are prescribed from the Model for OZone And Related chemical Tracers version 4 (MOZART-4) dataset (Emmons et al., 2010), in which the biogenic emissions of isoprene and monoterpenes are based on the Model for Emissions of Gases and Aerosols Emissions from Nature (MEGAN) (Guenther et al., 2006). The MEIC emission provides anthropogenic sources of the five VOC categories and the mapping table for lumping the MEIC VOC species to MOZART refers to Li et al. (2014). Since the MEIC does not provide the biogenic sources of isoprene and monoterpenes, which are much larger than their natural counterparts, we make their total emissions from anthropogenic and natural sources the same as those in the AR5 emission.

**Line 140-142, Did the factor of 1.4 also apply to the natural biomass burning emissions to derive SOAG?**

Yes, it does. When deriving SOAG emission, all anthropogenic and biomass burning sectors are added up and multiplied by 1.4 to convert OC (carbon mass) to POM (organic mass).

**Line 147, Is there particle number emissions in the AR5 emission inventories?**

No, the AR5 emission inventory only provides aerosol mass emissions. The number emission in both AR5 and MEIC emissions are calculated from the mass concentration as described in the text. To be clearer, the paragraph is rewritten as

follows:

"The number emission fluxes in both AR5 and MEIC are calculated from the mass fluxes in a consistent way. The mass to number conversion is based on $E_{number} = E_{mass}/\left(\frac{\pi}{6}\rho D_v{}^3\right)$, where $D_v$ is the volume-mean emitted diameter and ρ is the aerosol particle density (Liu et al., 2012). Since the MEIC emission does not provide mass emissions from agricultural waste burning, waste treatment, forest fire, grass fire and continuous volcanoes, we use the number fluxes from the AR5 emission for these sectors."

**Line 170, The simulated ADRE is a "all-sky" value while the observation-deriving ADRE in Line 187 is "clear-sky". Need to state the discrepancy.**

The cloud-screening method (clear-sky) in observations neglects the effect of aerosol above and below clouds. Absorption of reflected solar radiation by absorbing aerosol above clouds should result in larger radiative warming effect. The radiative cooling by scattering aerosol below clouds is not as strong as in clear sky due to less sunlight penetrating the clouds. Both of the two factors will results in lower (more negative at TOA) radiative effect in clear-sky than that in all-sky. Therefore, the observation-derived ADRE in clear-sky provides an upper-limit of the model-estimated all-sky ADRE at TOA.

**Line 211, It might be more appropriate to show concentration results (Section 3.2) before AOD results (Section 3.3).**

Thank you for the suggestion. It is more conventional to show surface concentration before AOD. Nevertheless, we show AOD first since we would like to address the question of whether the underestimated AOD by models is improved by using MEIC emission. Also, since satellite retrieved AOD provides the spatial distribution to be compared with the model results, we think it might be acceptable to show AOD before concentration results.

**Line 240, why not show the results by aerosol components?**

Following the referee's comment, we added the AOD by aerosol components (in the appendix). See the following figure.

[Figure]

[Figure]

Figure S5. The seasonal variation of longitudinal averaged (100$^{o}$E-124$^{o}$E) AOD at 550 nm by aerosol components (dust, sulphate, BC, POM, and SOA from top to bottom) simulated by CAM5-MAM3 using the MEIC emission (left column) and the AR5 emission (the right column).

**Line 249, "while the observed maximum extends further north" According to Figure 5, it seems that the observed AOD maximum is in the south of the simulated maximum. Please rewrite this sentence. Line 250-251 have the same problem.**

Following the referee's comment, we rewrite the sentences as "The model simulates a maximum between 35°N and 40°N in early summer (from May to July) which is to the north of the observed AOD maximum.".

**Line 249, "This model maximum is mostly due to dust aerosol …" It seems that the maximum AOD in the earlier summer is about half less than the observation, and the maximum AOD is from dust. Thus, the anthropogenic AOD is quiet low comparing with MODIS. That is, CAM5 model heavily underestimates AOD in China and this cannot be explained by emissions alone. Besides to the emissions, the author need to mention other causes (e.g. missing nitrate, particle size distribution, aerosol hygroscopic growth, etc.) that account for the large AOD bias.**

We agree with the referee that other causes beside emission also account for the large AOD bias. We mentioned the plausible causes and future effort to be made in Line 258-262 and the summary section (Line 443-449). We rewrite the sentences as "This model maximum is mostly due to dust aerosol transported from the west, while the satellite retrievals do not show such strong dust emission and transport. Since the dust emissions are the same in the simulations using the MEIC and AR5 emissions, the difference of the modeled AOD maxima between 35°N and 40°N is mainly due to anthropogenic aerosols. We notice that the maximum AOD in the satellite retrieval occurs around 30°N in early summer (May to July), as oppose to 35°N to 40°N as simulated by the model. The observed AOD maximum complies with the $SO_2$ emission maximum in early summer around 30°N (Figure S6). Therefore, this AOD maximum is heavily underestimated by the model. Since the uncertainty of

SO₂ emission is relative low ($\pm$12%), this underestimation cannot be explained by emission alone. Other causes (e.g. missing nitrate, particle size distribution, aerosol hygroscopic growth, etc.) in the model are more responsible." "

[Figure]

Figure S6. The seasonal variation of longitudinal averaged SO₂ emission from MEIC [g S/m²/year].

**Line 329-330, "The sulfate concentrations in northern China are characterized by the summer maximums" In Figure 9, at northern cities, the minimum wintertime sulfate in observations are susceptible. As the observations were collected from literature measurements that were carried out in different periods, the observations in summer and winter may not be comparable. The comparison uncertainty should be admitted.**

The summer maximums of the surface concentration of sulfate at northern cities in 2009 and 2010 were observed by some independent researches (Zhao et al., 2013; Zhang et al., 2013; Geng et al., 2015). It is possible that SO₂ to sulfate conversion is efficient in the summer due to high temperature and relative humidity. However, there were also observations showing opposite seasonal variations with sulfate concentration peaking in the winter during other periods (Zhang et al., 2012). The contrasting results in the observations could be due to that the observations were

carried out in different periods, as suggested by the referee. We have added this comparison uncertainty in the revised manuscript.

**Line 343-349, in my opinion, gas-phase oxidation of SO2 is not the main pathway for sulfate production. Aqueous oxidation in droplet/cloud water should be more important. At 35°-40°N, the maximum sulfate difference between MEIC and AR5 in summer is also due to the high ambient humidity. Besides, if CAM5 can capture the wintertime high concentrations, the largest sulfate burden difference could be in winter than in summer.**

We totally agree with the referee. We added the seasonal variation of relative humidity in Figure 11(f). One thing to notice is that aqueous-phase oxidation occurs only in cloud droplets in the model. The aqueous-phase oxidation of $SO_2$ by $NO_2$ on solution aerosols is not modeled by CAM5-MAM3, which is important in China (Wang et al., 2016). If we consider the oxidation by $NO_2$ on solution aerosols in CAM5, the contribution of aqueous-phase chemistry could be more important.

[Figure]

**Revised Figure 11.**

**Technical notes**

**If possible, mark the data range (i.e. Min, Max) in Figure 7 and 9.**

Data range (one standard deviation) in Figure 7 and Figure 9 are marked. The standard deviations of the surface concentrations are provided in only three locations (Beijing, Zhengzhou, and Guangzhou) by the literatures. We add them to Figure 9.

[revised manuscript text omitted]

---

## Referee Report (RR2)

This study attempts to estimate the performance of a newly developed emission inventory, i.e., the Multi-resolution Emission Inventory for China (MEIC) in predicting aerosols by comparing observations and simulations with the default IPCC AR5 emission inventory. The authors showed that the MEIC-driven CAM5 simulations underestimated aerosol pollution over eastern China but improved the prediction of magnitudes and seasonality of sulfate, primary organic aerosol, and black carbon (BC), when comparing to the AR5-driven one. Also, their simulations indicated that the changes in aerosol radiative forcing were significant due to the difference between the two emission inventories. This work is useful for improvement of emission inventory in China and can be published, provided that the following issues have been adequately addressed.

(1) Typically, the accuracy in global chemical transport model simulations depends on emission inventory, meteorology, and chemistry. The key features in the aerosol chemistry in China are related to very efficient secondary formation (Guo et al., *Proc. Natl. Acad. Sci. USA* **111**, 17373, 2014; Zhang et al., *Chem. Rev.* **115**, 3803, 2015). Specifically, the efficient secondary aerosol processes include aerosol nucleation and rapid growth under favorable conditions (Zhang et al., *Chem. Rev.* **112**, 1957, 2012; Qiu et al., *Phys. Chem. Chem. Phys.* **15**, 5738, 2013). It would be necessary for the authors to clearly state how those processes were accounted for in their chemistry module.

 (2) The modeling setups related to meteorology constrain need to be detailed and some interpretations for the modeling results should be more accurately stated. When using "constrained meteorology" mode to run the two primary simulations driven with MEIC and AR5 emission inventories (p6, lines 139-141), besides winds, have the authors also nudged temperature and moisture, which could be crucial for the gas- and aqueous-phase chemistry? From Figures 9 and 11, it seems both temperature and moisture are nudged because the temperature and relative humidity fields are identical in the two simulations. However, according to the discussions in section 3.3 about aerosol direct radiative forcing, the temperature as well as moisture fields for the two simulations should be different from each other because the surface cooling and atmospheric heating due to aerosols in the two simulations are different. How to interpret this conflict?

3) My other concern was tied up with the first one. Did the version of CAM5 model used in this study take care of the aerosol-meteorology interactions? Aerosol impacts on meteorological fields could be significant, which might further affect the aerosol pollution condition in the lower troposphere. Also, aerosol-cloud interactions might modify temperature and moisture profiles and precipitation (Wang et al., *Atmos. Chem. Phys.* **11**, 12421, 2011), leading to potential feedback on the atmospheric chemistry. Aerosol radiative effects induced by black carbon (BC) or other aerosol components could stabilize boundary layer and thus reduce the height of boundary layer, tending to exacerbate aerosol pollution near ground (Wang et al., *Atmos. Environ.* **81**, 713, 2013). A particular important aspect is the aging of BC, which considerably enhances light absorption (Khalizov et al., *J. Phys. Chem.* **113**, 1066, 2009; Peng et al., *Proc.*

*Natl. Acad. Sci. USA* **113**, 4266, 2016). Obviously, the aerosol-meteorology interactions cannot be ruled out when the authors attributing the source of the discrepancy between simulations and observations and the difference between the two simulations.

5) P7, lines 200-203: what's the reason to attribute the simulated AOD maximums between $35^o$N and $40^o$N to the transported dust aerosol? Note that in Figure 11h, the winds over the corresponding region in early summer are quite small, which does not support the long transportation of dust aerosol here. Could it be the efficient formation of sulfate over this region at this time period, based on the relatively high concentrations of sulfate (Figure 8) and relatively high gas- and aqueous-phase production of sulfate (Figure 9) at northern China sites like Chengde or Beijing?

6) I doubt the accuracy of the statement on p9, lines 254-255, saying that the MEIC emission inventory improves BC simulation relative to the case of AR5. if only looking at the BC scattering plot in Figure 7, the dots for MEIC simulation are loosely scattered in the plot comparing to AR5 case, actually suggesting that the prediction of BC by MEIC has larger uncertainty than AR5 case.

7) Since the authors employed a global climate model, it would be necessary to consider the potential impacts of climate changes on pollution conditions in China (Wu et al., *Sci. China: Earth Sci*. **59**, 1–16, 2016).

**Technical corrections**

1) Why there are two green lines (it's supposed to be only one) in Figure 9?

2) Figure 3 caption for (e) panel: "haft" should be "half".

3) P10, lines 273: add refs to support the statement of "this feature is commonly seen for many climate models".

---

## Author Response (AR2)

**Authors' response to the referees' comments on the revised manuscript (acp-2016-802)**

We thank the two referees' comments and suggestions for improving our manuscript. After we received the comments, we realized that we did not elaborate the contribution of our work to the scientific community well enough. Sometimes statement in the manuscripts leads to misunderstanding. Therefore, we made major revision to the manuscript. We have changed the title, analyzed the results in depth and in detail, revised the abstract and conclusions, and rewrote the statements to avoid misunderstanding. After having done all these, we believe the manuscript is now improved. Here we list the point-by-point reply to the two referees' comments. Referee's comments are in black and our replies are in blue.

**Reply to Referee #1**

Comments to the Revision (acp-2016-082):

I give a "Fair" for the Scientific Significance in the review report, because a single model simulation based on two emission inventories does not give too many impressive results. However, the manuscript adds a timely document for discussion of aerosol model uncertainty in China. The author has well responded to all the comments. Additional two notes are present below. I think the revised manuscript is ready for further consideration of acceptance.

Additional two minor notes:

(1) The revision has a new section 4 of discussing the decadal ADRE which is not mentioned in the abstract.

In the latest revision, we added the following statement in the abstract: " The differences of ADREs by using MEIC and AR5 emissions are larger than the decadal changes of the modeled ADREs, indicating the uncertainty of the emission inventories."

(2) Line 366-367, "… which is commonly seen for many climate models (personal communication with Prof. Hong Liao from Nanjing University of Information Science & Technology). " This is not a formal citation and is useless for reader. Please remove it.

This citation is now removed.

**Reply to Referee #3**

Suggestions for revision or reasons for rejection (will be published if the paper is accepted for final publication)

I do not recommend this manuscript be published based on the following reasons:

(1) I cannot find any significant scientific findings except the authors used a new emission inventory, MEIC.

It is our responsibility to elaborate the scientific significance of our work clearly, which we did not do very well. In the revised manuscript, we improved our statement of the significance and contribution of this research, especially to the modeling community. As indicated by many previous studies (e.g., Liu et al.,

2012; Shindell et al., 2013), global climate models suffer from substantially low bias of aerosol loadings in East Asia, in particular, the rapidly developing region of eastern China. The low aerosol loading bias can have significant implications for anthropogenic aerosol radiative forcing and climate effects as assessed by the IPCC ARs. It also suggests that IPCC aerosol forcing and climate effects could be much underestimated due to the large aerosol bias in China. The bias in anthropogenic aerosol emission is hypothesized to be one of the leading-order reasons.

Our research is of scientific value since we quantified how and how much the application of a new technology-based inventory developed by Chinese scientists to replace the widely used IPCC AR5 emission inventory can change the resultant aerosol concentrations and the radiative effect by a global climate model, which provides useful information for the modeling community to improve their models. We find that using the new emission inventory has distinct effect on primary and secondary aerosols, which implies that the modelers should follow different approaches (improving the emission inventory for primary aerosols or other aspects as well such as chemistry and aerosol dynamics for secondary aerosols) to improve the representation of these two types of aerosols. As such, we believe this research is of significant scientific value to the modeling community besides the sole use of a new emission inventory as claimed by the reviewer.

These points are now clearly stated in the introduction section and are reflected throughout the manuscript.

(2) The authors found the AOD simulations were improved by using MEIC, however, they also declared "MEIC emission leads to a better agreement with the observed surface concentrations of primary aerosols (i.e., primary organic carbon and black carbon) than the AR5 emission, while the seasonal variation of secondary aerosols (i.e., sulfate and secondary organic aerosol) depends less on the emission". The conclusions were not convincing, even as the authors stated in Section "3.1 Comparing MEIC and AR5 emission inventories", the emissions in eastern China had changed a lot, but could not be reflected by their secondary aerosol simulations. It is contradictory. The authors should analysis the modeling results in more detail to find reasonable explanations.

The above-mentioned statement in the abstract is not well written, which leads to confusion to the referee and the readers. The seasonal variation, instead of the annual average, of the surface concentration of secondary aerosols depends less on the emission (Fig. 8 in the latest revised version). The annual averaged surface concentrations of secondary aerosols, same as the primary aerosols, are improved as the result of the use of MEIC (Fig. 7 and Fig. 8), which is consistent with the improved AOD. The change of emission in eastern China from AR5 to MEIC **is reflected** in our simulated secondary aerosols. In addition to being impacted by emissions, secondary aerosols, which are produced in the atmosphere, are also impacted by the atmospheric processes. Thus secondary aerosols depends less on the emission, compared to the primary aerosols that are directly emitted.

The statement is now rewritten as "Seasonal variation of the MEIC emission leads to a better agreement with the observed seasonal variation of primary aerosols than the AR5 emission, but the concentrations are still underestimated. This implies that the atmospheric loadings of primary aerosols are closely related to the emission, which may still be underestimated over eastern China. In contrast, the seasonal variations of secondary aerosols depend more on aerosol processes (e.g., gas and aqueous phase production from precursor gases) that are associated with meteorological conditions and to a less extent on the emission. It indicates that the emissions of precursor gases for the secondary aerosols alone cannot explain the low bias in the model. Aerosol secondary production processes in CAM5 should also be revisited.".

We also add a new section 3.2 to discuss the distinct impact of emission and atmospheric processes on aerosol seasonal variation.

(3) The secondary aerosols (depends strongly on the emission changings as in eastern China) are very important for the ADRE calculations, especially the authors used internally mixed state within each mode. But we cannot find any modeling improvements or related explanations in the manuscript.

We agree with the referee that the secondary aerosols depend on the emission changings and are very important for the ADRE calculation. Our result shows that secondary aerosols (mainly sulfate and also SOA) dominate the total AOD (Table 1) and exert significant influence on the ADREs (Table 2). We are afraid that the referee may overlook the improvements of our simulated secondary aerosols and the related explanations. The use of a new emission inventory in our research does improve the modeling of the secondary aerosols (Fig. 7 and Fig. 8) and the ADRE (Table 2 and Fig. 15). We also explained the dominant aerosol processes in modeling the seasonal variation of the secondary aerosols (Line 274-284 in Section 3.2). We found that the simulated seasonal variation depends on the variation of gas- and aqueous-phase production rates in addition to the emission rates (Fig. 8 and Fig. 9). We also discussed missing mechanisms in our model (Line 285-294 and Line 418-427) and the meteorological impact (Line 295-320, Fig. 10 and Fig. 11). We believe that these investigations are, if not less, as equally important as making changes in the modeling mechanism. Any addition of modeling improvements should be based on the profound understanding of the performance of the current model, the physical mechanisms, as well as observations in the real atmosphere.

(4) The title is not "Impact of a new emission inventory on CAM5 simulations of aerosols and aerosol radiative effects in eastern China". Firstly, the ADRE is through the whole text, we cannot find semi-direct effects, "aerosol radiative effects" is not appropriate for the title. Secondly, I agree the other two reviewers' comments, no improvements for secondary aerosol formation were added in the simulations and the readers cannot find the secondary simulations be better than previous researches, "simulations of aerosols" is also not appropriate in the title.

We agree with the referee that this paper is about the aerosol direct radiative impact (ADRE). We do not focus on the semi-direct effect as this involves cloud

changes. For the second point, the word "simulation" may lead to an impression that our work is to improve the model treatments for the secondary aerosols. We change the title to "Emission or atmospheric processes? An attempt to attribute the source of large bias of aerosols in eastern China simulated by global climate models".

(5) The structure was not well, Section "3.1 Comparing MEIC and AR5 emission inventories" are not the authors' research results, others who are involved in simulations or emissions could get it.

This section is now moved to the supplementary.

In summary, we believe that our study will be an important contribution to the aerosol modeling community regarding the aerosol simulations in eastern China, pointing out the urgency of improving the emissions of aerosol and gas precursors in China and the atmospheric processes. We have made major revision in response to the referees' comments. We hope that the revised manuscript can be reconsidered for future acceptance.

Due to heavy modification of the manuscript, a mark-up manuscript version is almost impossible to read. Please refer to the previous version as attached below for comparison. Thanks!

[revised manuscript text omitted]

---

## Author Response (AR3)

We thank the reviewers for his/her helpful comments and suggestions on improving the manuscript. Below are our point-by-point responses. The reviewer's comments are in black and our responses are in blue.

Report #1

Submitted on 08 Aug 2017
Anonymous Referee #4

Review of Fan et al. (acp-2016-802)

In this study the authors examined the impact of a new emission inventory on aerosols and their radiative effects in eastern China using a global aerosol-climate model. Many global models suffer a low bias in simulating aerosols over the heavily polluted East Asia, and it is well known that aerosol emissions over there are highly uncertain. With the new technology-based multi-resolution emissions inventory for China, they found that low biases in AOD simulated with the widely used AR5 emissions are reduced and seasonal variations in primary aerosols are also improved, as evaluated against satellite retrievals and surface observations. These improvements in aerosols are found to have a significant impact on the regional aerosol direct radiative forcing. I have also read the comments on a previous version of the manuscript from two reviewers and the authors' responses. I agree with the reviewers on some of the good points and the "fair" scientific significance. However, I believe the authors have done a good job in addressing the comments and revising the manuscript, which has now become publishable. In particular, the new MEIC emissions files prepared for the CAM5 model, if made available to the community, would be a very useful contribution. Below I offer a few comments for the authors to consider before the final publication.

We thank the reviewer for the positive and encouraging comments.

Comments:

1) How does the MEIC emission inventory compare to the newly released CEDS (Community Emission Data System; Hoesly et al., 2017) for China? The CEDS dataset has been used in CAM5 by Yang et al. (2017) to study black carbon and its direct radiative forcing in China. I understand that the CEDS dataset probably wasn't available when this manuscript was first submitted, but since it is now released to the community and intended for use in CMIP6 simulations, the authors should include a discussion on this.

Thank you for mentioning the newly released CEDS dataset (Hoesly et al., 2017), which is a major step-forward for global emission inventory. According to Hoesly et al. (2017), CEDS follows a completely different approach than the country-level emission inventory, such as MEIC. A global default dataset was first compiled using activity data (e.g., energy consumption), emission factors, and emission inventories. Then a "mosaic" strategy is used to scale the default emission

estimates to authoritative country-level inventories. For China the CEDS dataset is scaled to MEIC for year 2008, 2010, and 2012 (Li et al., 2017) for most chemical species, except that BC and OC emissions are calculated using SPEW data (Bond et al., 2007). Gridded data are finally constructed using normalized spatial proxy (EDGAR gridded emission or HYDE population) distributions for each country. In terms of seasonality, the monthly fractions used in CEDS are from ECLIPSE project and are currently constant in time. Since our study is confined to eastern China, we do not consider smoothness with the surrounding area, MEIC is adequate and even better in seasonality than CEDS.

We added a section to compare MEIC and CEDS in the Supplement and added discussion on CEDS in the conclusions as follows:

"Recently, the Community Emission Data System (CEDS) is newly released and is intended for use in CMIP6 (Hoesly et al., 2017). The CEDS emission for eastern China is comparable with MEIC (see Section 3 in the Supplement) since CEDS is scaled to country-level inventories, i.e., MEIC for China (Li et al., 2017). Without improvements in the aerosol process, the similar low-bias over eastern China in CMIP5 GCMs are expected in CMIP6."

2) It is not very clear about the simulations performed in this study and results shown in some of the figures. Please clarify in section 2.1 as well as all of the relevant figure captions, including the time period of observations used for model evaluation. Why is a linear interpolation between year 2008 and 2010 needed to obtain MEIC emissions in 2009 (lines 163-164)? Please clarify.

We added the simulation time period (year 2009) in section 2.1 as well as in the captions of Figures 2, 6, 8, 12, 13, and 15. Since MEIC emissions are available in 2008 and 2010, we linearly interpolated between year 2008 and 2010 to obtain MEIC emissions in 2009. To clarify why we include observations in 2010, we change Line 163-164 from "To characterize the seasonal cycle of a full year, we extend our time selection of the observations from 2009 to 2010." to "Since the surface chemical composition data that covers a full year in 2009 is very limited, to compare at least one year's cycle of seasonal variation, we extend the range of time selection so that the observations are allowed to continue from 2009 to 2010. "

3) It is good to be precise, but I don't think it is really useful to keep so many significant figures (e.g., two digits after the decimal point) in some of the numbers in the results section, especially, for those numbers of percentage and/or with trailing zeros.

Done. Now all the percentages in the text and tables keep only one decimal place.

4) Supposedly, the AR5 $SO_2$ and primary aerosol emissions in Figure 3 are for anthropogenic sectors only (section 2.2), so there are no seasonal variations in $SO_2$ and BC, but why is there variation in POM? Please clarify.

We thank the reviewer for pointing out this issue for clarification. Actually,

there is a seasonal variation for BC, POM, and SO$_2$ emissions due to biomass burning and shipping. Volcanic source also contributes to seasonal variation of SO$_2$ emission. For fair comparison between MEIC and AR5, the MEIC emission in Figure 3 also include biomass burning and ships. We corrected Line 128-130 in Section 2.1: "The AR5 anthropogenic emissions of SO$_2$, BC, and POM do not have seasonal variations. With the inclusion of emissions from biomass burning and shipping for SO$_2$, BC, and POM, as well as volcanic source for SO$_2$, the total BC, POM, and SO2 emissions have seasonal variations in the AR5 emission inventory. However, we should note that the seasonal variation in the AR5 emissions is rather weak since anthropogenic emission dominates in eastern China."

Technical corrections:

Line 16: energy-statistics?
  Corrected.

Line 67: Is it "multi-scale" or "multi-resolution" for MEIC?
  MEIC stands for "Multi-resolution Emission Inventory for China" (Li et al., 2017). We have corrected the inconsistency throughout the manuscript.

Line 123: Did you actually change the model code to take the MEIC emissions? Or just prepare the emissions as input files for the model?
  We just prepare the emissions as input files for the model. We carried out a series of consistency checks to make sure, for example, that MEIC input works well with the "constrained meteorology", and that interpolation to the model resolution conserves total mass of the emitted species.

Line 199: Change "presents" to "presence"
  Corrected.

Line 343: Please make sure if R squared in Figure 8 is correlation coefficient.
R is correlation coefficient while R$^2$ depicts the portion of the variation of a variable explainable by the regression model. R-squared can also be used to measure the extent of scatter of data points. When a variable is more scatter, the distance between each data point and the regression line is large, and hence the R-squared value is small. We have change the terminology in the text and in the caption of Figure 7.

Figure 16 caption: Remove "change of".
  Removed.

In the nudged simulations, only horizontal winds are nudged toward the reanalysis with a relaxation time scale of 6 hours. This approach facilitates direct evaluation of model aerosols against observations at particular times and locations when the errors (and uncertainties associated with natural variability) in the modeled large-scale circulation is minimized. **Temperature and moisture are not nudged in this**

**study.** As evaluated in Zhang et al (2014), nudging temperature and moisture creates a large perturbation to the model state, resulting in unrealistic behavior for cloud and convection parameterizations because these parameterizations are calibrated based on the free-running model climate. Because winds are constrained, the advection of heat and moisture are constrained to some degree (when the difference in local temperature and moisture between two simulations is small), but local source and sink terms for atmospheric temperature and moisture are computed according to the model fast processes (e.g., cloud processes) and land processes (climatological sea surface temperatures are prescribed in the two simulations). The changes in atmospheric temperature and moisture can in turn influence the gas- and aqueous-phase chemistry and aerosols.

Our analysis shows that there are small differences in the temperature ($\Delta T < 1$ K) and moisture ($\Delta RH < 3\%$) between the MEIC run and the AR5 run in Figures 9 and 11 as shown in the figure below. However, the differences are almost indiscernible compared to seasonal variation, which is about 30-40 K in northern China and about 20 K in southern China (red curves in the second column of Figure 9). The temperature and moisture differences between the two simulations are indiscernible in Figure 9. The small changes in temperature and moisture reflect the differences in aerosol effects on meteorology through fast processes between the two aerosol emissions. Total impacts on temperature and moisture can be assessed by using a fully coupled, free-running earth system model, which is beyond the scope of this study (since we focus on the aerosol radiative forcing).

[Figure]

**Figure 1. Seasonal variation of the differences between the meteorological variables due to atmospheric and land fast processes introduced by aerosol differences between the MEIC and AR5 simulations in 10 locations in eastern China from north to south. From left to right: temperature (unit: K), relative humidity (unit: %), 10-m winds (unit:ms⁻¹), precipitataion (unit: mm/day).**

Next, we show that this temperature difference is reasonable. The magnitude of the temperature difference is the result of change of equilibrium state from AR5 to MEIC aerosol, which can be regarded as a radiative forcing ($\Delta F$), i.e., the energy change induced by different aerosol loadings between the two runs. The radiative forcing $\Delta F$ is calculated from difference between ADREs in the two simulations (-10.34 Wm⁻² for the AR5 run and -12.76 Wm⁻² for the MEIC run, see Table 2), which is -2.42 Wm⁻². We can obtain the change of surface temperature ($\Delta T_s$) by multiplying $\Delta F$ with the climate sensitivity, $\alpha$,

$$\Delta T_s = \alpha \Delta F$$

The climate sensitivity is estimated to be ~4 K with a doubling of $CO_2$ (3.7 $Wm^{-2}$) for CAM5. Therefore, the direct response of surface temperature, in the absence of the ocean feedbacks, is about 1 K.

We have included the above analysis in the supplementary, section 8. A detailed description on the nudging strategy is added in Section 2.1. We also add the following discussion in Section 3.2:

"In this study, changes in the aerosol radiative forcing will alter atmospheric temperature and moisture in the model, and can, in turn, influence gas- and aqueous-phase chemistry and aerosols. However, differences in temperature (< 1 K) and moisture (< 3%) are small enough compared to seasonal variations and therefore do not affect our finding on the impacts of emissions and atmospheric processes on aerosol burden. More discussion on the effect on aerosol-meteorological interactions is provided in Section 8 of the supplementary."

3) My other concern was tied up with the first one. Did the version of CAM5 model used in this study take care of the aerosol-meteorology interactions? Aerosol impacts on meteorological fields could be significant, which might further affect the aerosol pollution condition in the lower troposphere. Also, aerosol-cloud interactions might modify temperature and moisture profiles and precipitation (Wang et al., Atmos. Chem. Phys. 11, 12421, 2011), leading to potential feedback on the atmospheric chemistry. Aerosol radiative effects induced by black carbon (BC) or other aerosol components could stabilize boundary layer and thus reduce the height of boundary layer, tending to exacerbate aerosol pollution near ground (Wang et al., Atmos. Environ. 81, 713, 2013). A particular important aspect is the aging of BC, which considerably enhances light absorption (Khalizov et al., J. Phys. Chem. 113, 1066, 2009; Peng et al., Proc. Natl. Acad. Sci. USA 113, 4266, 2016). Obviously, the aerosol-meteorology interactions cannot be ruled out when the authors attributing the source of the discrepancy between simulations and observations and the difference between the two simulations.

Yes, the model version used in this study takes care of the aerosol-meteorology interactions. As mentioned by the reviewer, the aerosol-meteorology interactions introduce feedbacks that contribute to the discrepancy between simulations and observations and the difference between the two simulations. We note that the objective of this study is to quantify the aerosol radiative forcing (not aerosol effects) in China with using the two emissions. To minimize the effect of aerosol-meteorology interactions we use the "constrained meteorology" configuration and prescribed SST in this study. To quantify the feedbacks of aerosol-meteorology interactions, we will use a free-running and fully coupled atmosphere-ocean model. This is beyond the scope of this research and will be our future study.

5) P7, lines 200-203: what's the reason to attribute the simulated AOD maximums between 35°N and 40°N to the transported dust aerosol? Note that in Figure 11h, the

winds over the corresponding region in early summer are quite small, which does not support the long transportation of dust aerosol here. Could it be the efficient formation of sulfate over this region at this time period, based on the relatively high concentrations of sulfate (Figure 8) and relatively high gas- and aqueous-phase production of sulfate (Figure 9) at northern China sites like Chengde or Beijing?

We looked at the seasonal variation of AOD due to various aerosol species and verify the summer dust peak in Beijing, Xianghe, and Xinglong (See the Figure 5S below). We also noticed a summer dust peak in upper wind SACOL site. Therefore, the AOD maximums between 35°N and 40°N are mostly linked to the dust emission or transport. Although the 10-m winds in early summer are quite small in Figure 11 (for eastern China), dust aerosol can be lifted up to an altitude as high as 4-6 km in the Gobi and other Asian deserts. Transport at these altitudes is much easier than in the surface layer. However, we agree with the reviewer that sulfate is also a large contributor the maximum at northern China sites like Beijing, as shown in the Figure below, especially in July and August, when the simulated dust aerosol is at minimum.

[Figure]

**Figure S5. Seasonal variations of monthly mean AODs by aerosol species at 12 AERONET sites simulated by CAM5 using the MEIC emission inventory.**

6) I doubt the accuracy of the statement on p9, lines 254-255, saying that the MEIC emission inventory improves BC simulation relative to the case of AR5. if only looking at the BC scattering plot in Figure 7, the dots for MEIC simulation are loosely

scattered in the plot comparing to AR5 case, actually suggesting that the prediction of BC by MEIC has larger uncertainty than AR5 case.

It looks that the MEIC BC is more scattered than the AR5 BC compared with observations. The extent of scattering is measured by coefficient of determination ($R^2$), which is simply the square of correlation coefficient (R). R is calculated by the following equation

$$Correl(X,Y) = \frac{\sum (x-\bar{x})(y-\bar{y})}{\sqrt{\sum (x-\bar{x})^2 \sum (y-\bar{y})^2}}$$

We used NCAR Command Language (NCL) function "escorc" to calculate the Pearson linear correlation coefficient. To make sure we correctly computes the correlation, we use the statistic software "stata" and "excel" to calculate the value again. We also manually compute the correlation. They all give the same result. So we are confident that the number is correct. Why do MEIC BC dots look more scattering? It maybe due to several dots marked within green circles in the Figure below. There are not many of these dots but makes the dots visually more scattering. We notice that these dots are actually close to the 2:1 line, even the 1:1 line. So they are not bad simulation.

[Figure]

7) Since the authors employed a global climate model, it would be necessary to consider the potential impacts of climate changes on pollution conditions in China (Wu et al., Sci. China: Earth Sci. 59, 1–16, 2016).

We thank the reviewer for raising the good point. A predominant climatic phenomenon in China is East Asian monsoon, and thus the impacts of monsoon variability on air pollution have gained a lot of attentions (Wu et al., 2016). Monsoon and aerosol is a two-way interplay. The value of a global climate model not only resides in assessing the magnitude of the impact, but also in helping us understand individual mechanisms in the chain of interactions. Our current study aims to examine one of them, i.e., from emission to atmospheric processes. A series of 30-year runs

with the global climate model will be conducted to study the impacts of climate changes on aerosol pollution in China. We have added a discussion in the conclusion section of the revised manuscript.

Technical corrections

Why there are two green lines (it's supposed to be only one) in Figure 9?
The two green lines are for MEIC and AR5 runs.

Figure 3 caption for (e) panel: "haft" should be "half".
Corrected.

3) P10, lines 273: add refs to support the statement of "this feature is commonly seen for many climate models".
    This statement is from personal communication with Prof. Liao Hong at IAP (now at NUIST).

**References**

[revised manuscript text omitted]

Tianyi Fan 9/8/17 2:00 PM
已删除: 7.95
Tianyi Fan 9/8/17 2:00 PM
已删除: 87
Tianyi Fan 9/8/17 2:01 PM
已删除: 38
Tianyi Fan 9/8/17 2:01 PM
已删除: 85
Tianyi Fan 9/8/17 2:01 PM
已删除: 1
Tianyi Fan 9/8/17 2:01 PM
已删除: 18
Tianyi Fan 9/8/17 2:01 PM
已删除: 37
Tianyi Fan 9/8/17 2:02 PM
已删除: 0
Tianyi Fan 9/8/17 2:02 PM
已删除: 16
Tianyi Fan 10/16/17 11:05 AM
已删除: 2

More detailed comparisons with observations at 12 AERONET sites are given in Fig. 5. The model simulations using both emission inventories generally underestimate AOD compared with AERONET and satellite observations. The magnitudes of the AODs simulated with the MEIC emission are higher than that with the AR5 emission. The two simulations feature similar seasonal variations, for example, summer maximums at the sites north of 35 °N (Beijing, Xianghe, Xinglong, and

265 SACOL). This is because the simulated AODs are dominated by sulphate and dust aerosols at these northern sites and by sulphate aerosol at the southern sites (Taihu, Hongkong, and etc.) in both simulations (see Fig. S5 and Fig. S6 in the Supplement). Sulfate AOD peaks in summer in both simulations. In addition to the maximum in spring, dust AODs at the northern sites have two maximums in summer and autumn, which are suspicious and need further examination. The observed seasonality of AOD at northern sites features a maximum in July and a lower AOD in June, while the modeled

270 AOD peaks in June that may due to overestimated dust aerosol. The model captures the seasonality of observed AOD in the downwind regions but underestimates the magnitude of AOD by a factor of 2-3. AODs at the sites in 20-30°N (Taiwan and Hong Kong sites) are featured by the summer minimums in both observations and model results due to the scavenging of aerosols by the summer monsoon precipitation. The MEIC emission has a notable impact on AOD in Hong_Kong_PolyU site in all seasons and only has a small impact in winter at Taiwan sites (NCU_Taiwan, EPA_Taiwan, and Chen-

275 Kung_Univ). The difference between the two emission inventories is not evident at Osaka and Shirahama.

The sensitivities of modeled AOD to the emission change between the two inventories are quite different for each aerosol species due to different aerosol refractive indexes (Table 1). 12.0% of POM emission difference results in 70.4% of the AOD difference. In contrast, 46.9% of the SOAG emission difference leads to only 17.4% of the AOD difference of SOA.

*Single scattering albedo (SSA)*

280 Figure 6 shows the modeled SSA using the MEIC and AR5 emissions and the comparison with the observations by AERONET. The modeled SSA at Beijing, Xianghe, and Xinglong agrees with the AERONET data in terms of the strong seasonal variations of lower SSA in winter and higher SSA in summer, indicating higher fractions of absorbing aerosols in winter. However, the modeled SSA is systematically lower than the AERONET data. This indicates the significant underestimation of light scattering aerosols (e.g., sulfate and POM). The SSA simulated with the MEIC emission is lower

285 than that using the AR5 emission by up to 0.05 in winter, which is consistent with the higher BC emission in the MEIC emission. The SSA simulated with the MEIC emission in Taihu is slightly higher than that with the AR5 emission throughout the year, which is consistent with the higher MEIC emission of sulfate. Outside Mainland China the modeled SSA agrees with AERONET data reasonably well at the Hong Kong, Taiwan and Japanese sites, although underestimations can be found in some months.

Tianyi Fan 10/16/17 11:06 AM
已删除: 3
Tianyi Fan 10/16/17 11:06 AM
已删除: 4
Tianyi Fan 9/8/17 2:03 PM
已删除: 4
Tianyi Fan 9/8/17 2:03 PM
已删除: 35
Tianyi Fan 9/8/17 2:03 PM
已删除: 88
Tianyi Fan 9/8/17 2:03 PM
已删除: 0

[revised manuscript text omitted]
., 2017). According to Hoesly et al. (2017), CEDS follows a completely different approach than the country-level emission inventory, such as MEIC. A global default dataset was first compiled using activity data (e.g., energy consumption), emission factors, and emission inventories. Then a "mosaic" strategy is used to scale the default emission estimates to authoritative country-level inventories. For China the CEDS dataset is scaled to MEIC for year 2008, 2010, and 2012 (Li et al., 2017) for most chemical species, except that BC and OC emissions are calculated using SPEW data (Bond et al., 2007). Gridded data are finally constructed using normalized spatial proxy (EDGAR gridded emission or HYDE population) distributions for each country. In terms of seasonality, the monthly fractions used in CEDS are from ECLIPSE project and do not change by year. Since our study is confined to eastern China, we do not consider smoothness with the surrounding area, MEIC is adequate and even better in seasonality than CEDS.

The overall spatial distributions of MEIC and CEDS are similar, but local difference exists between the two inventories (Figure S2). Seasonal variations also show similar trend for BC and OC (Figure S3). However, significant difference can be found for $SO_2$. The magnitude of $SO_2$ emission in MEIC is about 20% lower than that in CEDS. Compared with MEIC, CEDS $SO_2$ emission shows smoother seasonal cycle characterized by high emission rates in winter and low emission rates in the summer. By examining the emission rates by sectors, we find that all sectors in MEIC $SO_2$ emission are smaller than CEDS (Table S2). Particularly, the energy/power sector produced $SO_2$ in MEIC (4853.8 Gg S/year) is 32% lower than that in CEDS (7171.2 Gg S/year). We also noticed that BC and OC emissions in the MEIC power/energy sector (2.0 Gg C/year and 0.035 Gg C/year, respectively) are much smaller than BC and OC emission in CEDS (654.6 Gg C/year and 1115.9 Gg C/year, respectively). This could be due to the use of SPEW data (Bond et al., 2007) for BC and OC emissions in CEDS (Hoesly et al., 2017). Generally, the CEDS emission for eastern China is comparable with MEIC since CEDS is scaled to country-level inventories. Without improvements in the aerosol process, the similar low-bias over eastern China in CMIP5 GCMs are expected in CMIP6.

[Figure]

**Figure S2. The spatial distributions of the MEIC emission, the AR5 emission and their difference for (a)-(c) SO$_2$, (d)-(f) BC, (g)-(i) POM of year 2009 in China.**

[Figure]

**Figure S3. Seasonal variations of sulfate, BC, and POM in the MEIC emission and the CEDS emission in China for year 2009.**

**Table S2. Statistics of the anthropogenic emission of SO2 (Gg S/year), BC(Gg C/year), and OC(Gg C/year) in MEIC and CEDS inventories for year 2009 in China**

| Species/Sectors | | MEIC | CEDS |
|---|---|---|---|
| $SO_2$ | industry | 8273.5 | 9399.9 |
| | power | 4853.8 | 7171.2 |
| | residential | 1649.2 | 1950.0 |
| | transportation | 106.6 | 237.4 |
| | total | 14883.1 | 18758.4 |
| BC | industry | 560.8 | 277.7 |
| | power | 2.0 | 654.6 |
| | residential | 893.8 | 962.3 |
| | transportation | 293.5 | 245.0 |
| | total | 1750.1 | 2139.5 |
| OC | industry | 517.8 | 232.4 |
| | power | 0.035 | 1115.9 |
| | residential | 2743.9 | 2033.9 |
| | transportation | 108.9 | 107.0 |
| | total | 3370.6 | 3489.1 |

120 **4. Observations of aerosol chemical compositions in China**

[revised manuscript text omitted]

**6. Figures**

Tianyi Fan 10/16/17 11:29 AM
已删除: 5

135

[Figure]

140

[Figure]

**Figure S4.** The seasonal variation of longitudinal averaged ($100^oE$-$124^oE$) AOD at 550 nm by aerosol components (dust, sulphate, BC, POM, and SOA from top to bottom) simulated by CAM5-MAM3 using the MEIC emission (left column) and the AR5 emission (the right column).

Tianyi Fan 10/16/17 11:05 AM
已删除: 2

[Figure]

**Figure S5. Seasonal variations of monthly mean AODs by aerosol species at 12 AERONET sites simulated by CAM5 using the MEIC emission inventory.**

Tianyi Fan 10/16/17 11:05 AM
已删除: 3

[Figure]

150

Figure S6. Same as Figure S3 but using the AR5 emission inventory.

Tianyi Fan 10/16/17 11:06 AM
已删除: 4

[Figure]

**Figure S7. The seasonal variation of longitudinal averaged SO₂ emission from AR5 and MEIC [g S/m²/year].**

Tianyi Fan 10/16/17 11:08 AM
已删除: 5

155

**Figure S8. Seasonal variations of surface concentration of sulfate at three locations (Gucheng, Panyu, and Zhengzhou) from CAWNET from 2006 to 2007. For comparison, the observations near the three CAWNET locations in our study (Beijing, Guangzhou, Zhengzhou) from 2009 to 2010 are also shown in dots.**

Tianyi Fan 10/16/17 11:13 AM
已删除: 6

160

**7. Decadal trend of emission from 2002 to 2012 over eastern China**

165    Figure S7 shows the MEIC's SO$_2$, BC, and OC emission trends from 2002 to 2012 in eastern China. Since spatially-gridded MEIC emission data are only available for 2008, 2010, and 2012, we obtained the spatial distribution and seasonal variation of other years by scaling the spatial-temporal variation of the emission in 2008 with the annual mean emission rates in these years. The annual mean emission rates of each species (SO$_2$, BC, and OC) are estimated by the MEIC development team. Each species in different sectors (power, energy, residential, and transportation) has a different scaling factor. The annual

170    trends are consistent with other researches (Lu et al., 2011; Lei et al., 2009) although the absolute values are different. We use the MEIC estimations because the algorithm and the database of fuel usage (i.e., China Energy Statistical Yearbook), are the same basis as the MEIC emission that we used for 2009.

[Figure]

175    **Figure S9.** The change of emission rates of SO$_2$, BC, and POM from year 2002 to 2012 over eastern China.

**8. Aerosol-meteorological interaction in the fast processes**

In the nudged simulations, only horizontal winds are nudged toward the reanalysis with a relaxation time scale of 6 hours. This approach facilitates direct evaluation of model aerosols against observations at particular times and locations when the errors (and uncertainties associated with natural variability) in the modeled large-scale circulation is minimized.

180    **Temperature and moisture are not nudged in this study.** As evaluated in Zhang et al (2014), nudging temperature and moisture creates a large perturbation to the model state, resulting in unrealistic behaviour for cloud and convection parameterizations because these parameterizations are calibrated based on the free-running model climate. Because winds are constrained, the advection of heat and moisture are constrained to some degree (when the difference in local temperature and moisture between two simulations is small), but local source and sink terms for atmospheric temperature and moisture

185    are computed according to the model fast processes (e.g., cloud processes) and land processes (climatological sea surface

Tianyi Fan 10/16/17 11:29 AM
已删除: 5

Tianyi Fan 10/16/17 11:15 AM
已删除: 7

temperatures are prescribed in the two simulations). The changes in atmospheric temperature and moisture can in turn influence the gas- and aqueous-phase chemistry and aerosols.

Our analysis shows that there are small differences in the temperature ($\Delta T < 1$ K) and moisture ($\Delta RH < 3\%$) between the MEIC run and the AR5 run in Figures 9 and 11 as shown in the figure below. However, the differences are almost indiscernible compared to seasonal variation, which is about 30-40 K in northern China and about 20 K in southern China (red curves in the second column of Figure 9). The temperature and moisture differences between the two simulations are indiscernible in Figure 9. The small changes in temperature and moisture reflect the differences in aerosol effects on meteorology through fast processes between the two aerosol emissions. Total impacts on temperature and moisture can be assessed by using a fully coupled, free-running earth system model, which is beyond the scope of this study (since we focus on the aerosol radiative forcing).

Next, we show that this temperature difference is reasonable. The magnitude of the temperature difference is the result of change of equilibrium state from AR5 to MEIC aerosol, which can be regarded as a radiative forcing ($\Delta F$), i.e., the energy change induced by different aerosol loadings between the two runs. The radiative forcing $\Delta F$ is calculated from difference between ADREs in the two simulations (-10.34 Wm$^{-2}$ for the AR5 run and -12.76 Wm$^{-2}$ for the MEIC run, see Table 2), which is -2.42 Wm$^{-2}$. We can obtain the change of surface temperature ($\Delta T_s$) by multiplying $\Delta F$ with the climate sensitivity, $\alpha$,

$$\Delta T_s = \alpha \Delta F$$

The climate sensitivity is estimated to be ~4 K with a doubling of $CO_2$ (3.7 Wm$^{-2}$) for CAM5. Therefore, the direct response of surface temperature, in the absence of the ocean feedbacks, is about 1 K.

[Figure]

[Figure]

Figure S10. Seasonal variation of the differences between the meteorological variables due to atmospheric and land fast processes introduced by aerosol differences between the MEIC and AR5 simulations in 10 locations in eastern China from north to south. From left to right: temperature (unit: K), relative humidity (unit: %), 10-m winds (unit:ms[-1]), precipitataion (unit: mm/day).

---

## Author Response (AR4)

We thank the reviewer for reviewing the revised manuscript and providing his/her valuable comments. We appreciate his/her suggestions and made corresponding changes to the manuscript. Below are our point-by-point responses. The reviewer's comments are in black and our responses are in blue.

Report #1

Submitted on 6 Dec 2017 Anonymous Referee #5

The authors have addressed most of the issues raised in my initial review. On the other hand, I still feel a couple of aspects that remained un-answered in their revision. I made the main points concerning whether the version of CAM5 model used in this study took care of the aerosol-meteorology interactions? Specifically, the aerosol impacts on meteorological fields could be significant, which might further affect the aerosol pollution condition in the lower troposphere. Also, the aerosol-cloud interaction might modify temperature and moisture profiles and precipitation (Wang et al., Atmos. Chem. Phys. 11, 12421, 2011), leading to potential feedback on the atmospheric chemistry. The aerosol radiative effects induced by black carbon (BC) or other aerosol components could stabilize boundary layer and thus reduce the height of boundary layer, tending to exacerbate aerosol pollution near the ground (Wang et al., Atmos. Environ. 81, 713, 2013). A particular important aspect is the aging of BC, which considerably enhances light absorption (Khalizov et al., J. Phys. Chem. 113, 1066, 2009; Peng et al., Proc. Natl. Acad. Sci. USA 113, 4266, 2016). Obviously, the aerosol-meteorology interactions cannot be ruled out when the authors attributing the source of the discrepancy between simulations and observations and the difference between the two simulations. Those are critical issues regarding the validity of their conclusions and need to be carefully assessed in their study.

We thank the reviewer for pointing out the importance of the aerosol-meteorology interactions (Wang et al., 2011, 2013; Khalizov et al. 2009; Peng

et al., 2016). We now include a new section (Section 3.3) to discuss this aspect of our model results and re-emphasize the impacts in the conclusions.

In the nudged simulations, only horizontal winds are nudged toward the reanalysis. Temperature and moisture are not nudged in this study. We do find differences in the simulated meteorological fields using the two inventories compared with the seasonal variations (less than 1 K for the surface temperature and less than 3% for the relative humidity). Although the differences are small probably due to the prescribed wind field, the impact could be significant if a fully coupled, free-running CESM-CAM3 is used. Also, to better simulate the impact we suggest to improve the aerosol treatment of BC aging, aqueous-phase reaction, and even the boundary layer microphysics, model meteorology, cloud and resolutions. Obviously, aerosol-meteorology interaction could be a significant aspect of aerosol pollution in East Asia and deserves improved treatment in CESM-CAM3 in the future.

**Emission or atmospheric processes? An attempt to attribute the source of large bias of aerosols in eastern China simulated by global climate models**

Tianyi Fan1, Xiaohong Liu1,2, Po-Lun Ma3, Qiang Zhang4, Zhanqing Li1,5, Yiquan Jiang6, Fang Zhang1, 5 Chuanfeng Zhao1, Xin Yang1, Fang Wu1, Yuying Wang1

[revised manuscript text omitted]
             | AR5              | (MEIC-AR5)/AR5 | MEIC                                   | AR5                                    | Schulz et al.,             |
|------------|--------------|------------------|------------------|----------------|----------------------------------------|----------------------------------------|----------------------------|
|            |              | ADRE,            | ADRE,            | ADRE,          | NRE,                                   | NRE,                                   | [2006] NRE,                |
|            |              | Wm -2 | Wm -2 | %              | $Wm^{\text{-}2}\tau_{aer}^{\text{-}1}$ | $Wm^{\text{-2}}\tau_{aer}^{\text{-1}}$ | $Wm^{-2}\tau_{aer}{}^{-1}$ |
| TOA        | All aerosols | -5.02            | -4.11            | 22.3%          | -20.83                                 | -22.05                                 |                            |
|            | Sulfate      | -2.62            | -1.96            | 33.6%          | -31.77                                 | -33.91                                 | -19                        |
|            |              |                  |                  |                |                                        |                                        | (-32 to -10)               |
|            | BC           | 2.51             | 1.81             | 39.1%          | 100.52                                 | 99.64                                  | 153                        |
|            |              |                  |                  |                |                                        |                                        | (28 to 270)                |
|            | POM          | -1.38            | -0.94            | 47.2%          | -33.84                                 | -36.70                                 | -19                        |
|            |              |                  |                  |                |                                        |                                        | (-38 to -5)                |
| Surface    | All aerosols | -18.47           | -14.99           | 23.3%          | -72.5                                  | -76.06                                 |                            |
|            | Sulfate      | -3.40            | -2.58            | 31.7%          | -40.36                                 | -43.78                                 |                            |
|            | BC           | -5.73            | -4.40            | 30.4%          | -204.98                                | -211.71                                |                            |
|            | POM          | -2.72            | -1.78            | 52.5%          | -63.73                                 | -68.04                                 |                            |
| Atmosphere | All aerosols | 13.45            | 10.88            | 23.6%          | 51.67                                  | 54.01                                  |                            |
|            | Sulfate      | 0.79             | 0.62             | 26.0%          | 8.58                                   | 9.87                                   |                            |
|            | BC           | 8.25             | 6.21             | 32.9%          | 305.50                                 | 311.35                                 |                            |
|            | POM          | 1.33             | 0.84             | 58.4%          | 29.89                                  | 31.35                                  |                            |
|            |              |                  |                  |                |                                        |                                        |                            |

Figure 1. Seasonal variations of sulfate, BC, POM, and SOAG in the MEIC emission and the AR5 emission in China for year 2009.